



# Exploring ozone variability in the upper troposphere and lower stratosphere using dynamical coordinates

Luis F. Millán[1], Peter Hoor[2], Michaela I. Hegglin[3,4,5], Gloria L. Manney[6,7], Harald Boenisch[8], Paul Jeffery[9], Daniel Kunkel[2], Irina Petropavlovskikh[10], Hao Ye[4], Thierry Leblanc[11], and Kaley Walker[9]

[1]Jet Propulsion Laboratory, California Institute of Technology, Pasadena, California, USA

[2]Institute for Atmospheric Physics, University of Mainz, Mainz, Germany

[3]Institute of Energy and Climate Research, Stratosphere (IEK-7), Forschungszentrum Jülich, Jülich, Germany

[4]Department of Meteorology, University of Reading, Reading, UK

[5]Department of Atmospheric Physics, University of Wuppertal, Wuppertal, Germany

[6]NorthWest Research Associates, Socorro, New Mexico, USA

[7]New Mexico Institute of Mining and Technology, Socorro, New Mexico, USA

[8]Karlsruhe Institute of Technology, Institute of Meteorology and Climate Research, Karlsruhe, Germany

[9]Department of Physics, University of Toronto, Toronto, Canada

[10]Cooperative Institute for Research in Environmental Sciences, National Ocean and Atmospheric Administration, Boulder, Colorado, USA

[11]Jet Propulsion Laboratory, California Institute of Technology, Wrightwood, California, USA

**Correspondence:** lmillan@jpl.nasa.gov

**Abstract.**

Ozone trends in the upper troposphere/lower stratosphere (UTLS) remain highly uncertain because of sharp spatial gradients and large variability caused by competing transport, chemical, and mixing processes near the upper tropospheric jets and extra-tropical tropopause, as well as inhomogeneous spatially and temporally limited observations of the region. Subtropical jets and

the tropopause act as transport barriers, delineating boundaries between atmospheric regimes controlled by different processes; they can thus be used to separate data taken in those different regimes for numerous purposes, including trend assessment. As part of the Observed Composition Trends And Variability in the UTLS (OCTAV-UTLS) Stratosphere-troposphere Processes And their Role in Climate (SPARC) activity, we assess the effectiveness of several coordinate systems in segregating air into different atmospheric regimes. To achieve this, a comprehensive dynamical dataset is used to reference every measurement from

various observing systems to the locations of jets and tropopauses in different coordinates (e.g., altitude, pressure, potential temperature, latitude, and equivalent latitude). We assess which coordinate combinations are most useful for dividing the measurements into bins such that the data in each bin is affected by the same processes, thus minimizing the variability induced when combining measurements from different dynamical regimes, each characterized by different physical processes. Such bins will be particularly suitable for combining measurements with different sampling characteristics and for assessing

trends and attributing them to changing atmospheric dynamics.



## 1 Introduction

The distribution of ozone in the upper troposphere/lower stratosphere (UTLS) region is crucial for the Earth's radiation budget (e.g., Riese et al., 2012),and for modulating air quality near the Earth surface (e.g., Langford et al., 2015; Lin et al., 2015; Williams et al., 2019). Despite its importance, and the decades of satellite, aircraft, balloon-borne, and ground-based measure-
ments, confidence in the long-term ozone trends in the UTLS remains low (e.g., Harris et al., 2015; Steinbrecht et al., 2017; Petropavlovskikh et al., 2019; Szeląg et al., 2020; Godin-Beekmann et al., 2022). The difficulty in quantifying trends arises because the UTLS is a transition region between the ozone-poor troposphere and the ozone-rich stratosphere (Gettelman et al., 2011). UTLS ozone also exhibits large spatial and temporal variability driven primarily by variations in the UTLS jets and the tropopauses (e.g., Hegglin et al., 2009; Pan et al., 2009; Manney et al., 2011; Schwartz et al., 2015; Albers et al., 2018; Olsen
et al., 2019). Measurements available in this region are spatially and temporally limited, resulting in inhomogeneous sampling of this variability. Moreover, the tropopause and the jets act as dynamical barriers to mixing, accompanied by strong changes of static stability (e.g., Birner, 2004) or strong isentropic potential vorticity (PV) gradients (e.g., Kunz et al., 2011a; Manney et al., 2011). Both lead to strong ozone and tracer gradients at the tropopause (Kunz et al., 2011b; Hegglin et al., 2008). Thus, tropopause (e.g., Pan et al., 2004; Hoor et al., 2004; Hegglin et al., 2009) or jet-relative (e.g., Manney et al., 2011; Olsen
et al., 2019) coordinate systems have often been used to segregate air masses influenced by different dynamical processes (e.g., tropospheric versus stratospheric or poleward versus equatorward of the subtropical jet).

Another way of segregating air masses is by using coordinates that account for adiabatic conservation laws, namely PV - potential temperature ($\theta$) related coordinates. Rossby and smaller scale waves lead to meridional displacements of air parcels that are mostly adiabatic and largely reversible in nature. PV-$\theta$ coordinates leverage the meridional distortions of PV contours
as well as the movement of adiabatic parcels on surfaces of constant $\theta$ to account for these displacements (e.g., Hegglin et al., 2006). It is important to note that irreversible processes (diabatic processes such as radiative cooling or heating, turbulent mixing and stirring) modify PV on different timescales. These processes are associated with transport that leads to mixing and irreversible tracer exchange, likewise introducing ozone variability that cannot be accounted for by adiabatic coordinate transformations. Analyzing datasets in geometric coordinate systems (e.g., latitude / pressure grids) generally results in larger
binned variability, as these coordinates do not account for the variability caused by changes in the positions of the jets or the tropopauses, nor for wave-induced air parcel displacements.

As part of the Observed Composition Trends and Variability in the UTLS (OCTAV-UTLS) Stratosphere-troposphere Processes And their Role in Climate (SPARC) activity, in this study we analyze how well different coordinate systems separate ozone measurements taken in atmospheric regimes dominated by different processes. Coordinate systems that effectively
achieve this are expected to segregate observations into bins with reduced variability because measurements influenced by different (reversible) dynamical processes will not be averaged together. The datasets used include observations from the Aura Microwave Limb Sounder (MLS) and the Atmospheric Chemistry Experiment-Fourier Transform Spectrometer (ACE-FTS) satellite instruments, as well as high resolution measurements from aircraft (including those from various research campaigns





and the Civil Aircraft for the Regular Investigation of the atmosphere Based on an Instrument Container (CARIBIC-2)), lidars,
and ozonesondes.

Each data point from our observational datasets (see section 2.1) comes with temporal and geolocation information. The geolocation information includes longitude and latitude in the horizontal and either altitude or pressure (or both) in the vertical. While these basic coordinates are essential for measurement retrievals and data processing, dynamically-defined coordinates often facilitate interpretation of the data. Coordinate systems designed to show relationships to atmospheric phenomena are
typically established with reference to the specific phenomenon itself, such as tropopause relative coordinates. Conversely, dynamical coordinates such as potential temperature in the vertical or equivalent latitude (i.e., PV on isentropes) in the horizontal provide a framework (based on conservation laws for atmospheric motions) that aligns with the adiabatic movement of the air parcels.

Each of these coordinates remap the data with respect to different aspects of dynamics, transport, or location. Thus, the
coordinates that are most helpful to study geophysical and transport properties of the data may be different for different regions of interest. A key metric used to evaluate the impact of binning the data in each coordinate system is the binned variability. Depending on the coordinate system and its ability to account for tracer gradients at transport barriers between different air masses (e.g., at the tropopause or jet cores), the binning process can induce artificial variability on top of the inherent atmospheric variability (e.g., induced by non-conservative processes).

Hegglin et al. (2008) introduced the term "geophysical noise" to describe this enhanced variability when comparing datasets binned using tropopause-relative coordinates to those binned using altitude. This comparison revealed increased variability when the influence of the tropopause (and the tracer gradients associated with its location) was not accounted for, thereby highlighting the significance of geophysical variability. Since geophysical variability is an inherent property of the atmosphere, different representations of the data, that is, coordinate systems that segregate dissimilar air masses, can minimize its effects on
values grouped together in a bin, making it a useful metric for coordinate system comparison. We emphasize that neither the geophysical variability itself nor atmospheric trace gas variability can be removed or minimized by any means – and indeed it is exactly this variability and the mechanisms for it that we ultimately want to isolate and study.

The choice of coordinate can, however, facilitate combining measurements in each bin that are primarily affected by the same processes (thus reducing the variability in that bin) by accounting for transport history and / or the locations of transport
barriers and thus strong tracer gradients. In other words, process-related coordinates can reduce binned variability, highlighting a more realistic representation of trace gas variability, and thus helping to elucidate the physical processes controlling it in different regions. The goal of this study is to show the effect of different coordinate systems on the binned variability. To achieve this, we use a variety of observational datasets together with reanalysis data.

Several acronyms are used throughout this paper; all are defined the first time they appear in the text. However, to improve
readability, a list of acronyms is provided in Appendix A (Table A1).



## 2 Datasets and binning methodology

### 2.1 Datasets

In this study we use UTLS ozone observations from a diverse set of measurement techniques, in particular, from ozonesonde, lidar, aircraft, and satellite datasets. These datasets have vastly different precision, accuracy, and temporal and spatial coverage. Table 1 provides a summary of the key characteristics of the different measurement systems, while Figure 1 displays the sampling patterns. Further information for each dataset is presented below.

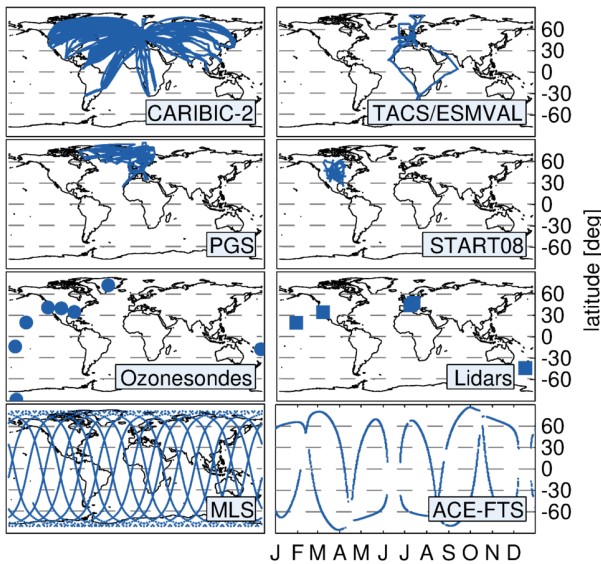

**Figure 1.** Sampling patters / locations of the ozone measurements used in this study. For the aircraft datasets (i.e., CARIBIC-2, TACTS/ESMVAL, PGS, and START08) we show all the sampling locations available during the 2005-2018 period. For ozonesondes and lidar datastes we display the site locations. For MLS and ACE-FTS we show representative daily and yearly sampling patterns respectively.

### 2.1.1 Satellite remote instruments

Satellite instruments operate remotely, enabling them to provide global coverage. They differ in their observation geometry and in the wavelengths they may use to remotely sense the atmosphere, which influence the measurement characteristics, accuracy, precision, and sampling. In this study, we focus on two satellite limb sounders, Aura MLS and ACE-FTS.

**Aura MLS**

Aura MLS was launched aboard the Aura satellite in July 2004 (Waters et al., 1999, 2006). The spacecraft flies in a 98° inclined near-polar, sun-synchronous orbit, with a 13:45 local time ascending (north-going) Equator-crossing time, at 705 km





altitude, that allows for observations from about 82°S to 82°N, each orbit. MLS uses heterodyne radiometers to observe
thermal emission from the atmospheric limb in spectral regions centered near 118, 190, 240, and 640 GHz, and 2.5 THz (i.e., at
wavelengths of 2.54, 1.58, 1.25, 0.47 and 0.12 mm). From these radiances temperature, trace gas concentrations, geopotential
height, and cloud ice are retrieved. MLS provides about 3500 profiles (per species) along the sub-orbital track every day, during
both daytime and nighttime. The MLS ozone vertical resolution in the UTLS is around $\sim 3\,\mathrm{km}$.

**ACE-FTS**

ACE-FTS was launched aboard the SciSat-1 spacecraft in August 2003 (Bernath et al., 2005). The spacecraft has a drifting orbit
at 650 km with an inclination of 74° that allows for observations from 85°S and 85°N. ACE-FTS profiles the atmosphere using
a solar occultation technique, measuring one sunrise and one sunset per orbit, resulting in approximately 15 sunrise and 15
sunset occultations per day. Global coverage is achieved over a period of three months (i.e., one season), with almost exactly
the same coverage year after year. ACE-FTS measures infrared sprectra between 750 and 4400 $\mathrm{cm^{-1}}$ at a high resolution
(0.02 $\mathrm{cm^{-1}}$) to derive volume mixing ratio profiles of over 50 atmospheric trace gas species and isotopologues (Boone et al.,
2005). These measurements achieve an effective resolution of around 1 km in the UTLS region due to vertical oversampling
(Hegglin et al., 2008).

In comparison with MLS, ACE-FTS has much lower sampling density and thus shows a seasonally varying sampling bias.
However, because of the very high signal-to-noise ratio of the solar occultation technique, ACE-FTS measurements are typi-
cally more precise than those from MLS.

### 2.1.2 Airborne in-situ instruments

Aircraft in-situ measurements for this study were typically made using chemiluminescence detectors and / or UV photometry.
In this study we use data from four campaigns:

- Stratosphere-Troposphere Analyses of Regional Transport (START08; Pan et al., 2010)

- Transport and Composition in the Upper Troposphere and Lower Stratosphere and Earth System Model Validation
  (TACTS/ESMVal; Müller et al., 2016)

- Polar Stratosphere in a Changing Climate (POLSTRACC; Oelhaf et al., 2019) campaign, operated with two other
  projects, the Investigation of the Life cycle of gravity waves (GW-LCYCLE) and Seasonality of Air mass transport
  and origin in the Lowermost Stratosphere (SALSA), known collectively as the PGS mission

- In-service Aircraft for a Global Observing System (IAGOS) Civil Aircraft for the Regular Investigation of the atmosphere
  Based on an Instrument Container (CARIBIC; Brenninkmeijer et al., 1999, 2007)

Typical random errors for the ozone measurements in these campaigns are smaller than 1% (e.g., Zahn et al., 2012). In
comparison to satellite instruments, in-situ measurements on aircraft generally have limited temporal and spatial coverage
globally, as shown in Figure 1. However, CARIBIC-2 aircraft operate at cruising altitudes of 10–13 km, near the climatological





location of the extratropical tropopause. The high temporal and horizontal sampling of CARIBIC-2 provides a very detailed
view of the tropopause and a very long time series (starting in 1997). In contrast, the other aircraft missions studied here,
START08, PGS, and TACTS/ESMVal, have more limited regional and temporal coverage, but provide more extensive vertical
coverage of the UTLS, making them ideal for process-oriented studies. Thus the set of all aircraft datasets used here provides
complementary views of the UTLS.

**2.1.3    Lidars**

This study uses data from several ground-based ozone differential absorption lidars (DIAL; Mégie et al. (1977)). Different
wavelengths are used for tropospheric (Hartley band: 266 nm–300 nm) and stratospheric ozone (Higgins band: 300 nm–360 nm)
to ensure adequate sensitivity to the drastically different ozone concentrations in the two regions. Stratospheric lidar measure-
ments used here are taken at Table Mountain, Mauna Loa, Haute-Provence, Hohenpeissenberg, and Lauder; tropospheric lidar
measurements are from Table Mountain and Haute-Provence (see Figure 1).

Because of the wavelength dependence, stratospheric ozone lidars only operate at night, while tropospheric ozone lidars
operate at any time of day (with limited signal-to-noise ratio during daytime). In this study, only nighttime data are used to
keep consistency between the tropospheric and stratospheric lidar datasets. Instruments operate for any duration from a few
minutes to numerous days (sometimes weeks) without interruption, typically recording 1–5 profiles a week at 5–20% relative
uncertainty in the UTLS. Most lidars achieve high vertical resolution, on the order of less than 1 km). Temporal and vertical
resolution can be tuned to achieve specific uncertainty requirements (Leblanc et al., 2016a, b). The characteristics of the lidars
used in this study are given in Table 1.

In comparison with satellite instruments, lidars can capture the temporal evolution of vertical ozone profiles over a given
location with relatively high vertical resolution and accuracy, but the geographical coverage is limited by the actual number of
instrument-locations.

**2.1.4    Ozonesondes**

The ozonesonde profiles used in this study (see Table 1 for details) are from balloons launched at eight stations (Summit,
Greenland; Trinidad Head, USA; Boulder, USA; Hunstville, USA; Hilo, USA; PagoPago, American Samoa; Suva, Fiji; and
Amundsen-Scott South Pole, Antarctica). Boulder, Hilo, and Trinidad Head stations have weekly ozonesonde launches, while
American Samoa and Fiji launch ozonesondes only twice a month, with occasional gaps in the time series. The sampling at
the South Pole station is typically weekly to bi-weekly, except during the ozone depletion season (September–October) when
sampling can be as frequent as every other day to map the rate of the ozone decline in the lower stratosphere (Johnson et al.,
2023). Since around 2001 (depending on the station), the data are collected with 1 Hz frequency, yielding a vertical resolution
between 5 to 300 m.

In this study, ozonesondes were gridded to 100 m to reduce computing power when calculating the dynamical diagnostics
(see Section 2.2). Lower stratospheric uncertainties of ozonesondes are about ±4–6% while in the upper troposphere they are





around ±5% in the tropics and around ±20% in mid-latitudes (e.g., Smit et al., 2007; Sterling et al., 2018; Tarasick et al., 2021; Smit and Thompson). The ozonesonde records have been homogenized to remove instrumental steps (Sterling et al., 2018).

Note that Stauffer et al. (2020) identified an instrument artifact that has caused total column ozone measurements from some
stations to drop by 3–7%, including Hilo, Fiji, and American Samoa. Subsequently, Stauffer et al. (2022) found that these drop-offs may be related to changes in the pump efficiency. These drop-offs were typically limited to pressures above ∼50 hPa, so the results shown here should be mostly unaffected. Further, in this study, we investigate the variability which should likewise be unaffected by these drop-offs.

In comparison with satellite instruments, ozonesondes, similar to lidars, can capture the temporal evolution of vertical ozone
profiles over a given location with high vertical resolution and accuracy, albeit with spatial coverage limited by the number of launch stations.

## 2.2 Method

### 2.2.1 Jet and tropopause characterization

To conduct a comprehensive analysis of the effects different coordinate systems can have on the variability of these ozone
datasets, supplementary information regarding the atmospheric dynamical conditions that affect them is essential. In the context of transport-relevant coordinates sought here, the information used in this study is potential temperature, equivalent latitude (the latitude that would enclose the same area between it and the pole as each isentropic potential vorticity contour), subtropical jet locations (derived from wind speeds), and tropopause locations at each measurement time and location. These dynamical fields were computed using the JEt and Tropopause Products for Analysis and Characterization (JETPAC) algorithms, which
are described in detail by Manney et al. (2011, 2014, 2017, 2021b), and Manney and Hegglin (2018). A complete overview of the latest JETPAC configuration used here is given by Millán et al. (2023).

In short, JETPAC provides potential temperature, equivalent latitude, dynamical (PV-based) and World Meteorological Organization (WMO, temperature gradient) tropopause locations and conditions, as well as the locations and dynamical characteristics of the UTLS jets for each of the measurement locations of the disparate datasets used here. JETPAC computes
these fields from reanalysis datasets, in this case the Modern Era Retrospective-analysis for Research and Applications, version 2 (MERRA-2; Gelaro et al., 2017). MERRA-2 provides meteorological fields at 3-hour intervals, on a 0.625° by 0.5° latitude/longitude grid with 72 hybrid $\sigma$-pressure levels between the surface and 0.01 hPa. The UTLS vertical spacing is about 1.2 km. MERRA-2 products have been extensively evaluated and found to be well-suited for UTLS studies (Manney et al., 2017, 2021a, c; Manney and Hegglin, 2018; Homeyer et al., 2021; Fujiwara et al., 2022; Tegtmeier et al., 2022).

By using the same algorithms and the same reanalysis fields for all datasets, we ensure that the derived dynamical conditions are consistent throughout the diverse datasets used in this study. This consistency facilitates the examination of these datasets with varying sampling characteristics, uncertainties, and resolutions in a unified dynamical framework. This framework allows us to explore the impact of different dynamical coordinate systems such as equivalent latitude, potential temperature,





tropopause and jet relative coordinates, as well as to compare with conventional coordinates such as latitude, altitude, and
pressure.

### 2.2.2   Coordinate mapping

We examine the effects of different coordinate systems on the representation of geophysical variability in UTLS ozone through production of climatologies from the datasets outlined in Section 2.1. Because the variability in these climatologies is also influenced by sampling and measurement characteristics, the use of multiple datasets allows exploration of the commonalities
among differences in climatologies as a function of coordinate system for each instrument. Any common changes between coordinate systems are assumed to result from a change in the representation of the effects of geophysical variability.

In this study, we focus on 3-month climatological periods, using data spanning 2005 through 2018. In particular, we focus on December-January-February (DJF) climatologies constructed for this 14-year period, to investigate the perspective given by using different coordinate systems. Results for the June-July-August (JJA) period are provided in the appendix for further
reference. We highlight these seasons to focus on the periods where the subtropical jet is predominant in the northern hemisphere (DJF) as well as in the southern hemisphere (JJA) (e.g., Spensberger and Spengler, 2020; Manney et al., 2014). Results for March-April-May (MAM) and September-October-November (SON) were analyzed but are not shown.

All information for dynamical coordinates (e.g., equivalent latitude, jet and tropopause characteristics) used in the construction of the climatologies is calculated using JETPAC. In the vertical, data are binned from their native pressure or altitude onto
uniform vertical grids, using either altitude, pressure, or potential temperature, with bounds of each chosen to span approximately the same vertical range within the UTLS. Figure 2 illustrates the redistribution of ozone across these three coordinates, when plotted versus latitude as the horizontal coordinate.

Additional vertical coordinates are constructed by setting the altitude or potential temperature in reference to the tropopause or the subtropical jet (STJ) core. In this study, three tropopause definitions were considered: the WMO-defined lapse rate
tropopause, the dynamically-defined 2 potential vorticity unit (PVU), and the 4.5 PVU tropopause. In total this results in eleven vertical grids, as outlined in Table 2. An example of these relative coordinates is illustrated in Figure 3, which shows ozone plotted as a function of latitude and potential temperature relative to the three tropopauses used in this study. The bounds of the vertical coordinate grids were chosen to minimize contributions from the lower troposphere and middle stratosphere to the UTLS climatologies.

In the horizontal, data are binned onto grids using either geographic latitude, equivalent latitude, or STJ-relative latitude (STJ-L). Each of these coordinates uses a 5° spacing but the geographic and equivalent latitude grids span 90° N to 90° S, while the STJ-L grid spans 30° equatorward to 60° poleward of the jet core. The influence of the jets is limited to a smaller latitudinal range than what is employed here. However, the 30-degree to 60-degree range allow us to compare against other coordinate systems in the most straightforward manner. These horizontal coordinates are summarized in Table 3, and illustrated
in Figure 4 using potential temperature as the vertical coordinate. Note than when referring to STJ-L we divide the data into hemispheres resulting in the two subpanels per dataset as shown in the bottom row of Figure 4. This separation by hemisphere is also performed when referring to the subtropical jet core in the vertical.



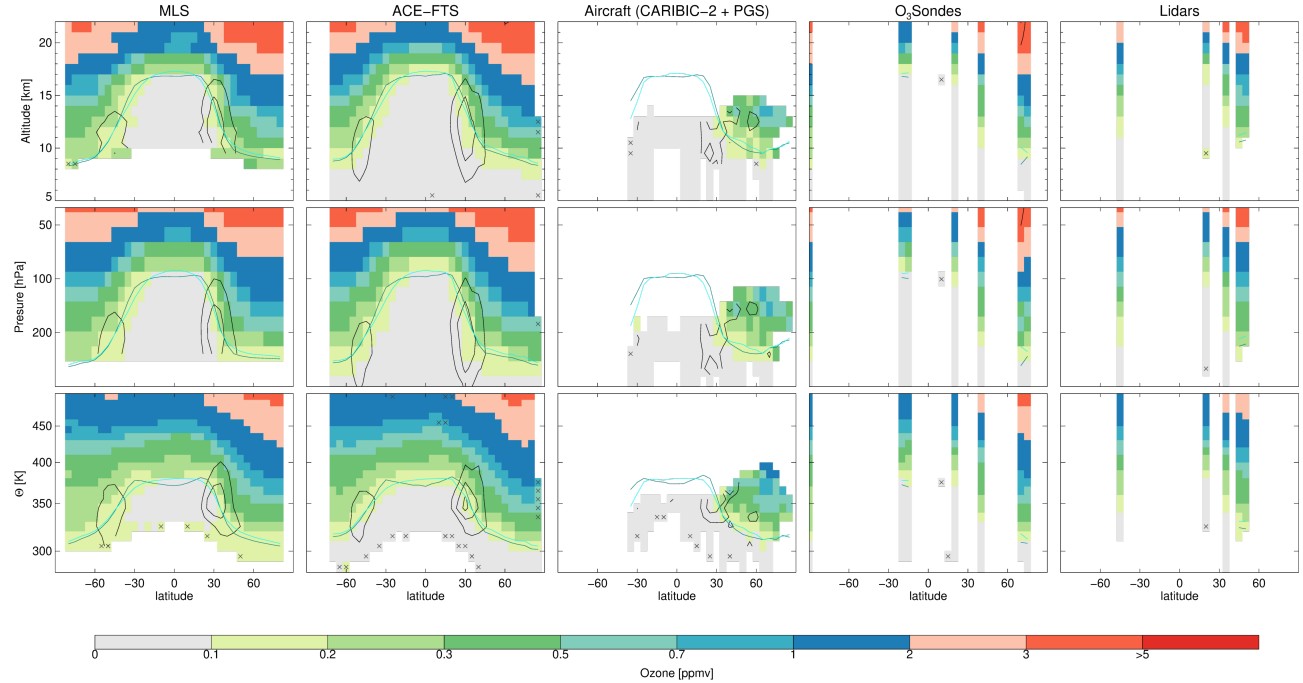

**Figure 2.** DJF (2005-2018) mean ozone distributions for MLS, ACE-FTS, aircraft, ozonesondes, and lidars as a function of latitude and altitude, pressure, or potential temperature. Cyan lines show the 4.5 PVU dynamical tropopause, and teal lines the WMO (thermal) tropopause. The black contours show wind speed values of 30, 40, and 50 m s$^{-1}$. Note that differences in the wind representation in comparison with MLS suggest sampling biases. Crosses indicate bins where there are less than 10 measurements.

The effect of the dynamical remapping using equivalent latitude or jet-based coordinates is most noticeable for the ozonesonde and lidar datasets. These observations are made near fixed geographical latitudes, but for different dynamical conditions (e.g., south of the STJ or north of the STJ, different tropopause altitudes, etc). The use of dynamical coordinates bins the data according to dynamical regimes, thus accounting for the dynamical conditions over time at a fixed location. It therefore expands their "condition-space" coverage to span much of the globe.

For each coordinate bin (spanning 5° in the horizontal coordinate and the vertical spacing outlined Table 2), we quantified the variability using the relative standard deviation, $RSTD$, given by

$$RSTD = \frac{\sigma}{\bar{x}} \tag{1}$$

where $\bar{x}$ is the mean VMR, and $\sigma$ is the standard deviation of the bin. The RSTD is used to evaluate the variability of the climatologies, as it provides a measure of spatial variance that is scaled and thus independent of the magnitude of the mean concentration within each coordinate bin, enabling effective comparisons across the UTLS.





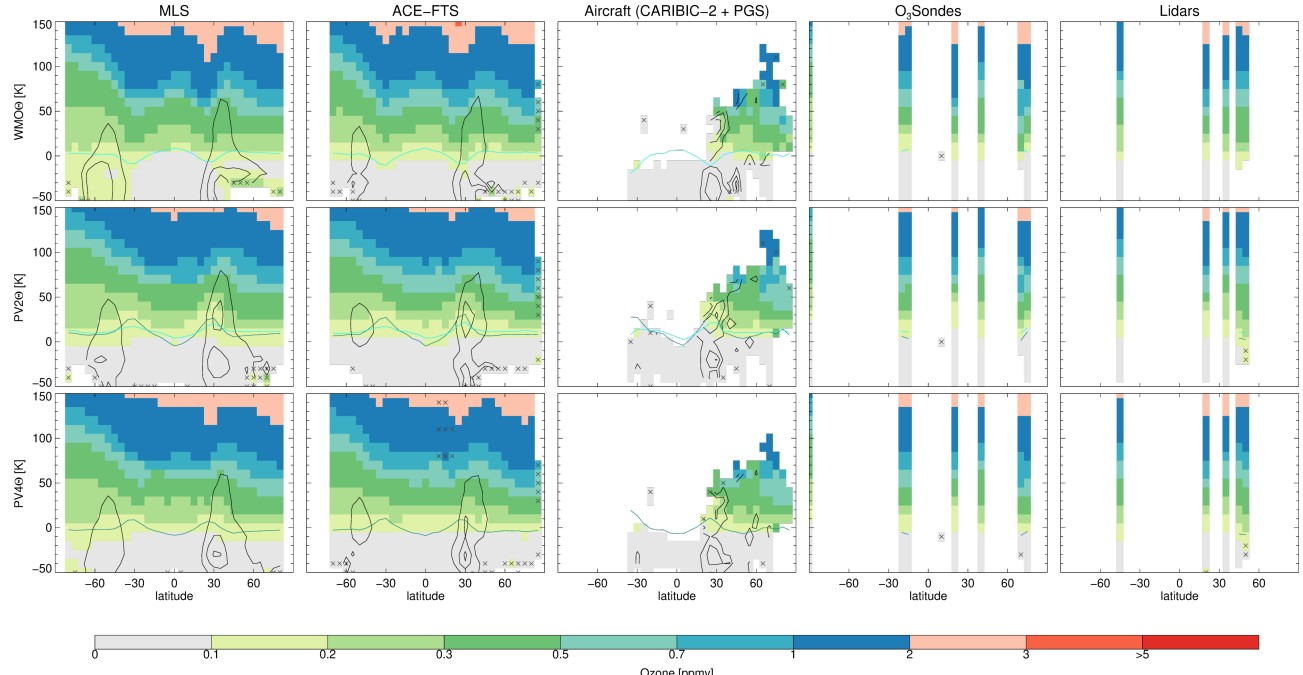

**Figure 3.** As Figure 2, but the vertical coordinates represent potential temperature difference with respect to the tropopause defined by the WMO criteria, 2 PVU or 4.5 PVU threshold.

## 3   Results

Before conducting a comparison involving all 33 coordinate systems, we assess the RSTD and underlying properties for the coordinate systems illustrated in Figures 2 through 4. The RSTD equivalents to those figures are shown in Figures 5 through 7. It is important to note that the aircraft, ozonesonde, and lidar datasets have much sparser coverage, particularly for the latter two, in latitude-based coordinates than ACE-FTS and MLS. Additionally, ACE-FTS and lidar observations are limited to clear-sky conditions due to their inability to penetrate most clouds. MLS has the coarsest vertical resolution, causing smearing of both observations and the variance (Livesey et al., 2020). The aircraft measurements are mostly limited to flight levels but allow for the detection of more variability in the measurement region due to the high temporal sampling. By examining these diverse datasets, the impacts of each individual limitation in resolution or sampling can be assessed and ozone variability characteristics that are robust across all datasets can be identified. As a reference, Figure A1 showcases the number of measurements per bin available for each observation system and for several coordinate systems used in this study.

Figure 5 shows the influence on the relative standard deviation of using different traditional vertical coordinates versus latitude. The tropopause region can be clearly identified as a region of high ozone variability in all five datasets. In altitude and pressure coordinates, the variability associated with this feature extends well into the troposphere, particularly for MLS as a consequence of its coarser vertical resolution. However, when employing potential temperature, which effectively captures



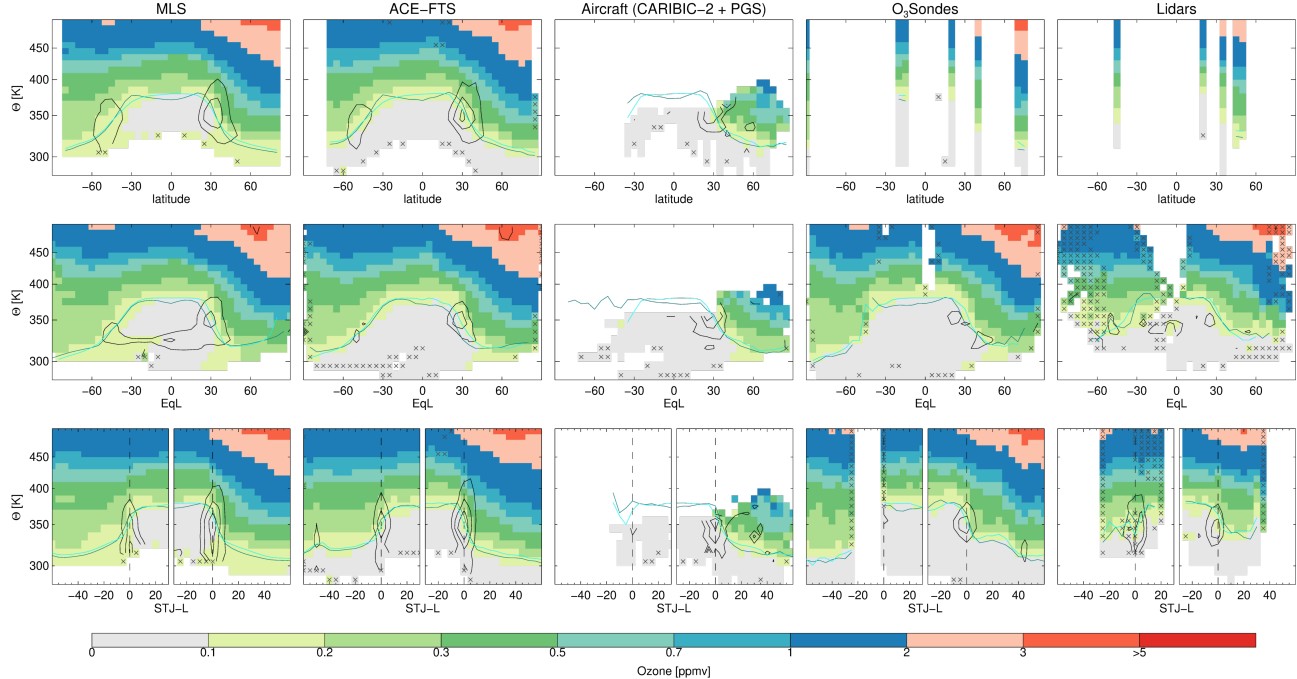

**Figure 4.** As Figure 2, but the vertical coordinate is potential temperature and the horizontal coordinates in the three rows are latitude, equivalent latitude, and distance in latitude from the STJ (i.e., STJ-L).

rapid quasi-isentropic transport and accounts for vertical displacements of the adiabats in altitude or pressure coordinates, a
decrease in the vertical extent of this high binned variability (high RSTD values) becomes apparent. This effect is particularly evident in the MLS, ACE-FTS, and aircraft datasets but can be inferred in the ozonesonde and lidar plots as well. Thus the potential temperature vertical coordinate helps account for some of the geophysical variability seen when binning the data in altitude or pressure.

Moreover, MLS and ACE-FTS display particularly high RSTD values around the northern STJ, which constitutes a stronger
transport barrier in DJF compared to summer. Specifically, the STJ separates tropical from midlatitude air, leading to intense ozone gradients near the jet location. Variability in this region manifests itself as high RSTD values resulting from variations in the latitude of large ozone gradients. In altitude and pressure coordinates, the jet-associated variability mostly falls along the tropopause. However, when employing potential temperature, the jet-induced variability manifests more prominently as a distinct lobe of variability located mostly above the STJ. Overall, as a function of latitude, potential temperature not only
reduces the overall binned variability, but also clarifies the structure of two main sources of variability (i.e., tropopause and STJ variations), which cannot be separated when using altitude or pressure coordinates.

When changing the vertical coordinate to potential temperature relative to the tropopause(s), as shown in Figure 6, it is apparent that the effect of remapping to tropopause-referenced coordinates tends to agglomerate the variability in the bins





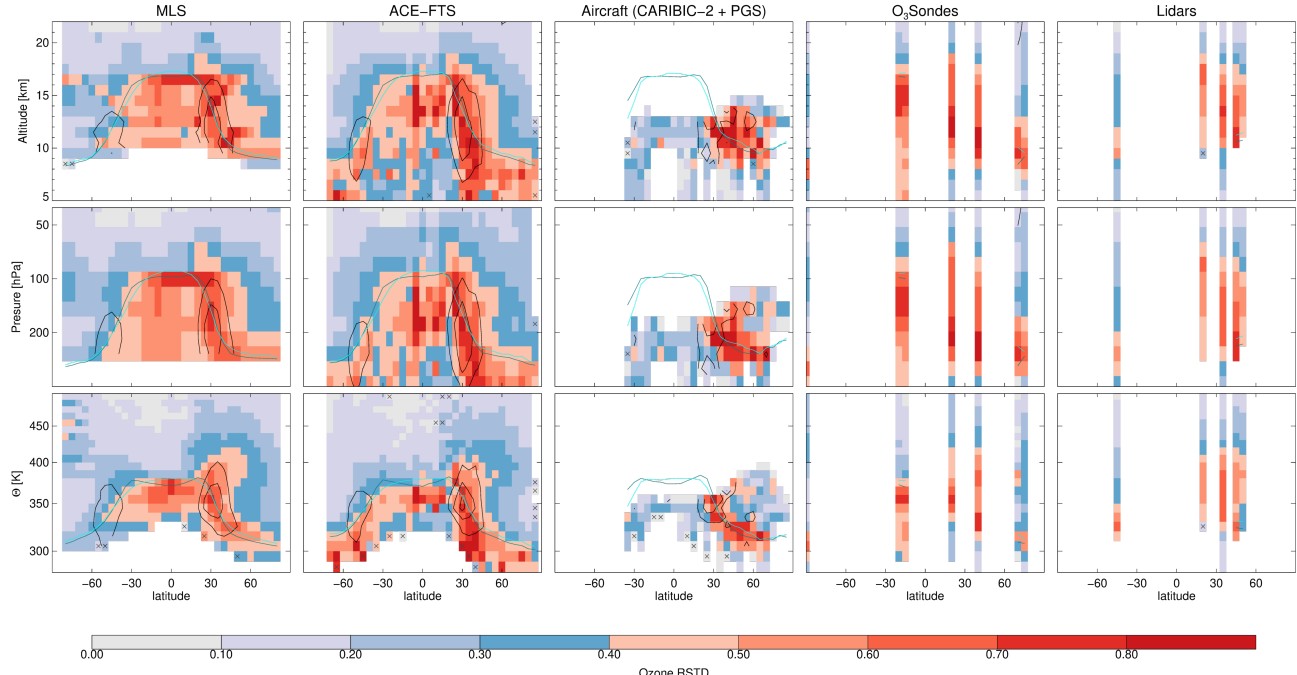

**Figure 5.** As in Figure 2, but displaying the ozone relative standard deviation.

along the transport barriers. In other words, where gradients are strong enough to make for substantial changes within one bin.

MLS and ACE-FTS display high RSTD lobes around 30° S and 30° N (also hinted by the lidars), which correspond to the regions where double tropopauses (e.g., Schwartz et al., 2015; Olsen et al., 2019) associated with tropospheric and stratospheric intrusions are preferentially found. In these regions, the choice of one reference surface leads to high variability in a fixed latitude frame. The location of the double tropopauses varies with latitude, longitude, and time. The binning process then mixes measurements within latitude bins where vertical profiles are taken relative to the lower (primary) and upper (secondary)

tropopause at different longitudes, resulting in the large RSTD at the jet location high into the lower stratosphere. Moreover, in between the primary and secondary tropopause, only accounting for the vertical distance relative to the tropopause fails to account for the presence of air masses with tropospheric and stratospheric origin that are quasi-horizontally (i.e., quasi-isentropically) advected between these levels, leading to high RSTD values.

By intercomparing the panels in Figure 6, it is evident the RSTD deviations are overall smaller when binning with respect to

either the 2 PVU (PV2$\theta$) or 4.5 PVU dynamical tropopause (PV4$\theta$) than when binning with respect to the WMO tropopause (WMO$\theta$). In particular, this is noticeable in the lobes of high RSTD around 30° S and 30° N seen in MLS and ACE-FTS and hinted at in the ozonesonde and lidar panels. Further, the RSTD also displays smaller values for the aircraft datasets in the extratropics, with the PV4$\theta$ coordinate generally accounting for variability better than the PV2$\theta$ coordinate. This is because the 2 PVU surface better represents the tropopause at mid and high latitudes (Hoor et al., 2004; Kunz et al., 2015), while higher PV





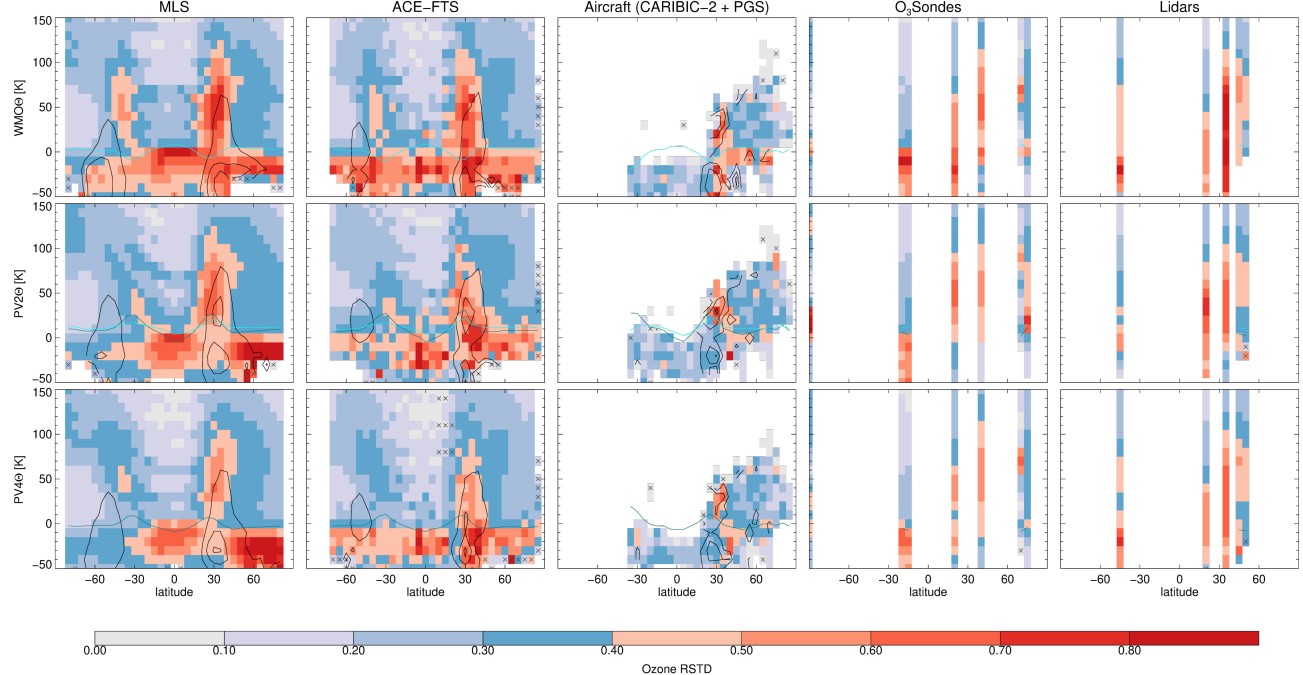

**Figure 6.** As Figure 3 but displaying the ozone relative standard deviation.

values best represent the tropopause for the subtropics (Kunz et al., 2015; Berthet et al., 2007). In general, Figure 6 suggests that dynamical tropopause based coordinates resolve the ozone gradients across the tropopause region better than the WMO tropopause based coordinate. Note that in comparison with the other datasets, MLS displays larger (smaller) RSTD values in the Northern (Southern) hemisphere extratropics in the tropopause-based coordinates, despite its coarse vertical resolution potentially failing to properly resolve the tropopause, which might be related to its better coverage of the region, i.e., MLS might sample more variability.

Figure 7 shows the influence on the RSTD of using different horizontal coordinates with potential temperature as the vertical coordinate. The similarity between binning in latitude and binning in STJ-referenced latitude in the MLS and ACE-FTS panels is striking, though the STJ-L panels do show variability along the STJ having a narrower peak (especially for ACE-FTS). This similarity likely arises from the relatively dense sampling of these datasets and the climatological averaging; it also likely arises partly from the fact that the jets have a strong influence on transport only in the region within about 20–30 degrees latitude of the jet, meaning that distributions are expected to be very similar away from the jets. A similar effect is seen for the aircraft data, albeit with slightly higher RSTD values in the extratropics, again suggesting that the limited latitude region of influence of the jets is an important factor.

For MLS, ACE-FTS, and the aircraft datasets, binning in equivalent latitude leads to a reduction in RSTD. For example, the lobe of high RSTD above the northern STJ in latitude and at zero degrees in the STJ-L coordinate system is greatly reduced



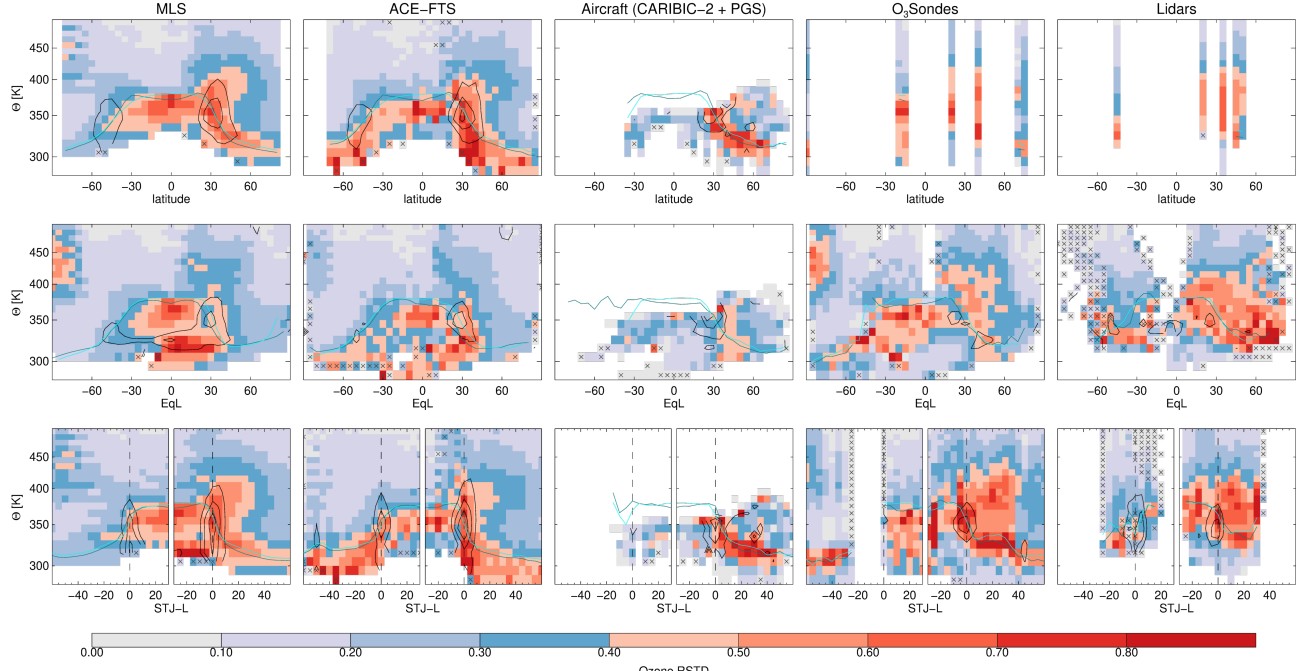

**Figure 7.** As Figure 4, but displaying the ozone relative standard deviation.

when binning the data using equivalent latitude. This is also evident in the ozonesonde and lidar datasets, which show higher RSTD values when binned in latitude with respect to the STJ compared to binning using equivalent latitude. The high RSTD values are greatly reduced when using equivalent latitude, which accounts for the different dynamical regimes and isentropic PV gradients away from the STJ. This illustrates that a portion of the variability is related to reversible processes, in this case, primarily the undulation of planetary and synoptic-scale waves. In contrast, binning with respect to STJ leads to pronounced RSTD values at the jet core location (i.e., zero degrees with respect to STJ) due to the strong ozone gradient across the jet, but further from the jet the variability is larger than observed with the other horizontal coordinates.

Overall Figure 7 indicates that all datasets benefit from the use of equivalent latitude. This coordinate implicitly includes the dynamical tropopause and accounts for dynamics on the typical time scale of planetary wave activity by accounting for the reversible part of the planetary wave induced air mass excursions on the mean, especially in the lower stratosphere. However, it is important to exercise caution when using equivalent latitude coordinates in the upper troposphere. Adiabatic PV conservation is violated, particularly by phase transitions of water, regional turbulence (especially near jet cores), radiative cooling in anticyclones (e.g., Zierl and Wirth, 1997) or above clouds (e.g., Kunkel et al., 2016), and the absence of connected circumpolar transport barrier (e.g., Manney et al., 2011; Pan et al., 2012; Jin et al., 2013; Kaluza et al., 2021). STJ relative coordinates are more appropriate for processes in the region immediately surrounding the jet and for studies where identifying and quantifying the strength and sharpness of the jet is critical.



## 4    Discussion

We now discuss the DJF RSTD for MLS (Figure 8), ozonesondes (Figure 9), and aircraft measurements (Figure 10) to illustrate the differences between the various coordinate systems. These datasets were chosen to exemplify satellite observations (MLS) with relatively coarse vertical and horizontal resolution but with global coverage, as well as examples for in-situ data with fine vertical resolution (ozonesondes) and horizontal resolution (aircraft) but with limited geographical coverage. The equivalent figures for ACE-FTS and lidars are shown in the appendix (Figures C1 and C2).

Despite their different sampling and data densities all datasets show broad areas of agreement (i.e., consistency in the change) when comparing the various binned coordinate systems in Figure 8, Figure 9, and Figure 10. Comparing the typically used vertical coordinates (altitude, pressure, and potential temperature) versus latitude, equivalent latitude, and latitude relative to STJ (top three rows in these figures), all datasets show a significant reduction of binned variability in the equivalent latitude-potential temperature coordinate system. For example, the binned variability directly at the extratropical tropopause (including the subtropics) is greatly reduced. Further, the lobes of variability above the northern STJ as seen in MLS (Figure 8) almost disappear in this coordinate system.

This result may not be too surprising since equivalent latitude-potential temperature constitutes a purely adiabatic coordinate system combining isentropes (i.e., adiabats) with PV, which is materially conserved for adiabatic and frictionless flow. Equivalent latitude facilitates identifying a reversible adiabatic transport component (which can be appropriately accounted for using suitable coordinates) and a non-adiabatic component related to irreversible mixing. The latter cannot be fully accounted for by coordinate mapping and constitutes part of natural atmospheric variability. In contrast, minimizing the impact of the former is contingent upon the selection of suitable coordinates.

Regarding tropopause relative coordinates, we subdivided the data into two sub-categories using geometrical altitude and potential temperature relative to the tropopause. Across all datasets, the use of a tropopause-relative altitude coordinates consistently results in larger binned variability than tropopause-relative potential temperature coordinates, regardless of the horizontal coordinate used. This highlights again the quasi-isentropic stratospheric distribution of ozone. Overall, the binned variability is smaller for all horizontal coordinates when using either the 2 PVU or the 4.5 PVU tropopause as a reference than when using the WMO tropopause. Further, the 2 PVU relative coordinate in general leads to higher binned variability in the subtropics than the 4.5 PVU coordinate but with very similar binned variability elsewhere. The enhanced tropical variability when using the 2 PVU tropopause is in line with the findings of Kunz et al. (2011a), which concluded that the subtropical tropopause is better represented by the ~4-5.5 PVU surfaces, depending on the season.

As discussed in section 3, double tropopauses associated with the STJ manifest as regions of enhanced ozone variability (around 30°S and 30°N) since a vertical coordinate with respect to the primary tropopause cannot account for mixing measurements taken relative to the lower (primary) and upper (secondary) tropopause, as well as, air masses reference with tropospheric and stratospheric origins that are quasi-horizontally advected between the primary and secondary tropopauses. These lobes of binned variability are somewhat reduced when using STJ-referenced latitude. However, away from the STJ core, the binned variability increases since the jet is a primary factor (that is, a transport barrier) in controlling the flow only in a narrow latitude




band around the jet core, and thus the flow away from this region is being better represented by a dynamical coordinate such as equivalent latitude.

Binning in an equivalent latitude - tropopause-referenced coordinate results in high binned variability near the south pole during DJF (and near the north pole during JJA, see appendix). In fact, it leads to larger binned variability than using latitude
or STJ latitude at all times. This is related to the thermal structure in the polar regions, where both WMO and dynamical tropopauses are often ill-defined and/or very broad (e.g., Bethan et al., 1996; Zängl and Hoinka, 2001; Wilcox et al., 2012).

Regarding vertical STJ coordinates, the data are again shown for both, coordinates relative to altitude and relative to potential temperature. As expected, across all datasets, the use of a STJ coordinates relative to altitude results in larger RSTD values than for a STJ coordinates relative to potential temperature, regardless of the horizontal coordinate used. Examining the RSTD
values across the coordinate systems which use STJ$\theta$ as the vertical coordinate, it is evident that the binned variability is minimized when using STJ-L as the horizontal coordinate. That is, referring to STJ both in the vertical and in the horizontal leads to the smallest binned variability within the STJ based coordinates.

All of the findings discussed in this section also hold for ACE-FTS and lidar datasets (see Figures C1 and C2), as well as for the other seasons (see Figures D1–D5 for JJA examples).
To further quantify the impact of using the various coordinate systems on variability, we use the binned climatological values of latitude and pressure in the different coordinate systems to remap the variability into the "traditional" latitude-pressure coordinate system. The accuracy of this remapping depends on how representative the climatological latitude and pressure values are in the different coordinate systems. Consequently, it only offers a broad overview of the impacts of these coordinate systems. The difference between the RSTD of the reference coordinate system ($RSTD_{ref}$) and that of each of the remapped
climatologies ($RSTD_{clim}$), is calculated as:

$$RSTD_{diff} = \frac{RSTD_{clim} - RSTD_{ref}}{RSTD_{ref}}, \tag{2}$$

which yields a direct metric for assessing the effect of these coordinate systems upon the binned variability.

Figure 11 displays the result of remapping and comparing the MLS DJF data. Notably, the use of equivalent latitude-potential temperature (EqL/$\theta$) coordinates leads to the most substantial reduction in binned variability across the upper troposphere and
lowermost stratosphere. Equivalent latitude effectively accounts for the reversible short-term variability at the extratropical tropopause, while potential temperature accounts for the vertical variability of isentropic surfaces and thus isentropic vertical displacement of air parcels. To highlight this further, Figure E1 displays the same MLS DJF data comparison using the climatological values of equivalent latitude and $\theta$, i.e., remapping into EqL/$\theta$. In all other coordinates, there is enhanced binned variability, except in small regions, emphasizing the global utility of this coordinate pairing.
Figure 11 also highlights that any tropopause based coordinate leads to reduced binned variability around the tropopause, consistent with the results of Hegglin et al. (e.g., 2009). It is important to note that, at greater distances from the respective tropopauses, tropopause coordinates in altitude tend to increase variability for all horizontal coordinates. This confirms earlier results by Hegglin et al. (2008) and indicates that using tropopause-based altitude coordinate systems may not be physically



meaningful farther away from the tropopauses. Similarly STJ coordinates lead to reduced binned variability only around the
STJ consistent with the results of Manney et al. (2011).

The reduction in variability observed in tropopause-based coordinates relative to potential temperature, especially in the winter hemisphere when the Brewer-Dobson circulation dominates vertical movement via advection, supports this interpretation (e.g., Hoor et al., 2004; Hegglin et al., 2006). Unlike altitude, potential temperature accounts for at least some of the large-scale adiabatic movements driven by the stratospheric circulation in the deeper stratosphere on shorter timescales (e.g.,
Harzer et al., 2023). Finally, some of this enhanced variability in all tropopause-based coordinates is generally further reduced when using latitude with respect to the STJ.

## 5   Summary

As part of the SPARC OCTAV-UTLS activity, we have mapped multi-platform ozone datasets into different coordinate systems to systematically evaluate the influence of these coordinates on binned climatological variability, unifying the disparate work
of numerous prior studies on individual coordinate system variability into the most complete assessment of this topic that we are aware of. Coordinate systems that do not consider transport barriers can induce artificial variability when binning across ozone gradients at transport barriers, increasing the binned variability. By comparing the relative standard deviation in different coordinate systems we evaluated the ability of each coordinate to account for variations arising from changes in the subtropical upper tropospheric jet, changes in tropopause height, and wave-induced air parcel displacements. We thus evaluated the ability
of each coordinate system to identify different regimes separated by transport barriers, and to group air parcels appropriately into those regimes.

We found that:

- Across all datasets the use of tropopause or subtropical jet vertical coordinates results in larger binned variability for altitude based coordinates compared to potential temperature based coordinates, regardless of the horizontal coordinate
used. This highlights the largely quasi-isentropic distribution of upper tropospheric and lower stratospheric ozone.

- Any tropopause based coordinate (compared to commonly used coordinates such as altitude and pressure) leads to reduced binned variability just around the tropopause, consistent with previous studies. However, higher variability is seen in tropopause-based coordinates at some distances from the respective tropopauses.

- The binned variability is smaller for all horizontal coordinates when using either the 2 PVU or the 4.5 PVU tropopause
as a reference than when using the WMO tropopause.

- STJ-relative latitude leads to somewhat reduced binned variability in a narrow latitude band around the STJ core; farther from the STJ, equivalent latitude better represents the air parcels' movement.



– The use of equivalent latitude-potential temperature coordinates leads to the most substantial reduction in binned variability across the UTLS through all datasets and all seasons. Because this coordinate system uses PV on isentropic surfaces, and PV is conserved for adiabatic frictionless flow, the transport of tracers follows this coordinate system.

We note that each coordinate system has its strengths and weaknesses, and thus different coordinate systems may be most effective for times / regions dominated by variability from different atmospheric process. In this study we identified coordinate systems that most help to reduce binned variability over broad regions in an effort to facilitate more robust UTLS composition trend analyses. A future OCTAV-UTLS study will evaluate the impact of using these coordinates that most reduce binned variability on quantification of long term ozone trends. Another study will analyze how differences in sampling patterns and resolution (both vertical and horizontal) can affect the representation of the datasets as well as the trend quantification.

*Data availability.* The ozone datasets used are available as follows:

– OzoneSondes: https://gml.noaa.gov/aftp/data/ozwv/Ozonesonde/

– Lidar: https://www-air.larc.nasa.gov/missions/ndacc/data.html

– START08: https://data.eol.ucar.edu/master_lists/generated/start08/

– TACTS/ESMVal: https://halo-db.pa.op.dlr.de/

– PGS: https://halo-db.pa.op.dlr.de/

– CARIBIC-1 and 2: https://www.caribic-atmospheric.com/Data.php

– ACE-FTS: http://www.ace.uwaterloo.ca

– ACE-FTS quality information: https://dataverse.scholarsportal.info/dataset.xhtml?persistentId=doi:10.5683/SP2/BC4ATC

– Aura MLS: https://disc.gsfc.nasa.gov/

For the dynamical diagnostics please contact Gloria L Manney (manney@nwra.com) or Luis F Millán (lmillan@jpl.nasa.gov).

*Author contributions.* A first draft of this manuscript was written by all coauthors during an ISSI workshop. LM rewrote that manuscript aiming for cohesion. All the co-authors commented on and edited the manuscript.

*Competing interests.* There are no competing interests.

*Acknowledgements.* This research was supported by the International Space Science Institute (ISSI) in Bern, through ISSI International Team project #509 (Understanding Satellite, Aircraft, Balloon, and Ground-Based Composition Trends: Using Dynamical Coordinates for Consistent Analysis of UTLS Composition). LFM's and TL's research was carried out at the Jet Propulsion Laboratory, California Institute



of Technology, under a contract with the National Aeronautics and Space Administration (80NM0018D0004). GLM was supported by sub-
contracts from JPL through the Microwave Limb Sounder (MLS) project (JPL Subcontract #1521127). PH and DK acknowldege support
by the German Science foundation (DFG) by the TRR 301 (Project 428312742). IP's research was supported by the NOAA Cooperative
Agreement with CIRES, NA17OAR4320101 and the NOAA Earth's Radiation Budget (ERB) project. We thank the JPL MLS team (espe-
cially Brian Knosp and Ryan Fuller) for data management and processing support, and William Daffer for work on early development of
JETPAC. MERRA-2 is an official product of the Global Modeling and Assimilation Office at NASA GSFC, funded by the NASA Modeling
Analysis and Prediction program. The Atmospheric Chemistry Experiment is a Canadian-led mission primarily supported by the CSA. The
ozonesonde data are supported by NOAA GML and NASA SHADOZ observational programs. The lidar data used in this publication were
obtained from T. Leblanc, W. Steinbrecht, S. Godin-Beekmann, and R. Querel as part of the Network for the Detection of Atmospheric
Composition Change (NDACC) and are available through the NDACC website www.ndacc.org.



**Table 1.** Dataset characteristics.

| | Name | Region | Timespan | Range | Techinque | References |
|---|---|---|---|---|---|---|
| Ozonesondes | Summit, Greenland, SUM | 72.6N 38.4W | 2005-2017 | 0-30[a] km | ECC[b] | Sterling et al. (2018) |
| | Trinidad Head, USA, THD | 41.0N 124.1W | 1997- | 0-30 km | ECC | Sterling et al. (2018); Stauffer et al. (2022) |
| | Boulder, USA, BLD | 39.9N 105.2W | 1967-1971, 1979- | 0-30 km | ECC | Sterling et al. (2018); Stauffer et al. (2022) |
| | Huntsville, USA, HVA | 34.7N 86.6W | 1999- | 0-30 km | ECC | Sterling et al. (2018); Stauffer et al. (2022) |
| | Hilo, USA, HIH | 19.7N 155.0W | 1982- | 0-30 km | ECC | Sterling et al. (2018); Stauffer et al. (2020) |
| | Tutuila, American Samoa, SMO | 14.2S 170.5W | 1986-1990, 1995- | 0-30 km | ECC | Sterling et al. (2018); Stauffer et al. (2020) |
| | Suva, Fiji, SUV | 18.0S 178.0E. | 1997- | 0-30 km | ECC | Sterling et al. (2018); Stauffer et al. (2020) |
| | South Pole, Antarctica, SPO | 89.9S 24.8W | 1967-1971, 1986- | 0-30 km | ECC | Johnson et al. (2023) |
| Lidars | Hohenpeissenberg, Germany, HOH | 47.8N 11.0E | 1978- | 10-50 km | Strat $O_3$ DIAL[c] | Steinbrecht et al. (2009) |
| | Obs. Haute Provence, France, OHP | 43.9N 5.7E | 1991- | 0-12 km | Trop $O_3$ DIAL | Ancellet et al. (1989) |
| | Obs. Haute Provence, France, OHP[d] | 43.9N 5.7E | 1985- | 10-45 km | Strato $O_3$ DIAL | Pelon et al. (1986) |
| | JPL Table Mountain Facility, USA, TMF | 34.4N 117.7W | 1999- | 0-23 km | Trop $O_3$ DIAL | McDermid et al. (2002) |
| | JPL Table Mountain Facility, USA, TMF[e] | 34.4N 117.7W | 1989- | 12-50 km - | Strat $O_3$ DIAL | McDermid et al. (1990) |
| | Mauna Loa, USA, MLO | 19.5N 155.5W | 1993- | 10-50 km | Strat $O_3$ DIAL | McDermid et al. (1995) |
| | Lauder, New Zealand, LAU | 45.0S 169.6E | 1994- | 10-50 km | Strat $O_3$ DIAL | Bernet et al. (2020) |
| Aircraft | CARIBIC-2 | N. Hemisphere | 2005-2020 | fligt level | CLD and UV pht[f] | Brenninkmeijer et al. (2007) |
| | START08 | Continental US | 2008 | fligt level | CLD and UV pht | Pan et al. (2010) |
| | TACTS/ESMVAL | Europe and Africa | 2012 | fligt level | CLD[g] | Müller et al. (2016) |
| | PGS | Arctic | 2015-2016 | fligt level | CLD[g] | Oelhaf et al. (2019) |
| Satellite | Aura MLS (v5) | 82S-82N[h] | 2004- | ~9-150km | Limb emission | Waters et al. (2006) |
| | ACE-FTS (v4.1/4.2) | 85S-85N[i] | 2004- | 5-95km | Solar occultation | Bernath et al. (2005) |

[a] For all ozonesondes, the highest altitude depends on the bursting point of the balloon

[b] Electrochemical Concentration

[c] DIfferential Absorption Lidar

[d] There are two different lidars at OHP, a stratospheric system (measuring since 1985) and tropospheric one (measuring since 1991).

[e] There are two different lidars at TMF, a stratospheric system (measuring since 1989) and tropospheric one (measuring since 1999).

[f] photometry

[g] These campaigns all used the FAIRO instrument (Zahn et al., 2012).

[h] daily, [i] seasonally





| Coordinate | Vertical range (resolution) |
|---|---|
| Altitude (A) | 5 km to 22 km (1 km) |
| Pressure (P) | 400 hPa to 40 hPa (12 levels per decade) |
| Potential temperature ($\theta$) | 250 K to 480 K (10 K) |
| Thermal tropopause-relative altitude (WMOA) | 5 km below to 5 km above the tropopause (1 km) |
| Thermal tropopause-relative potential temperature (WMO$\theta$) | 50 K below to 150 K above the tropopause (10 K) |
| 2-PVU dynamical tropopause-relative altitude (PV2A) | 5 km below to 5 km above the tropopause (1 km) |
| 2-PVU dynamical tropopause-relative potential temperature (PV2$\theta$) | 50 K below to 150 K above the tropopause (10 K) |
| 4.5-PVU dynamical tropopause-relative altitude (PV4A) | 5 km below to 5 km above the tropopause (1 km) |
| 4.5-PVU dynamical tropopause-relative potential temperature (PV4$\theta$) | 50 K below to 150 K above the tropopause (10 K) |
| STJ-relative altitude (STJA) | 5 km below to 5 km above the jet (1 km) |
| STJ-relative potential temperature (STJ$\theta$) | 50 K below to 150 K above the jet (10 K) |

**Table 2.** Vertical coordinate grids employed in this study, along with their vertical ranges and resolution.

| Coordinate | Horizontal range (resolution) |
|---|---|
| Geographic Latitude (Lat) | 90° N to 90° S (5°) |
| Equivalent Latitude (EqL) | 90° N to 90° S (5°) |
| STJ-relative latitude (STJ-L) | 30° equatorward to 60° poleward of STJ (5°) |

**Table 3.** Horizontal coordinate grids employed in this study, along with their ranges and resolution.





**Figure 8.** Overview of the MLS DJF (2005-2018) ozone relative standard deviation. Cyan lines show the 4.5 PVU dynamical tropopause, and teal lines the WMO (thermal) tropopause (dotted teal lines show the secondary thermal tropopause). The black contours show wind speed values of 30, 40, and 50 ms$^{-1}$.



**Figure 9.** As Figure 8, but displaying the ozonesonde relative standard deviation.





**Figure 10.** As Figure 8, but displaying the aircraft relative standard deviation.





**Figure 11.** Relative standard deviation changes in different coordinates in comparison to binning in latitude-pressure. Red colors indicate an increase in binned variability, while blue colors denote a reduction in binned variability.



# A   Acronyms and symbols used in this study





| | | | |
|---|---|---|---|
| A | Altitude | PV4A | 4.5-PVU dynamical tropopause-relative altitude |
| ACE-FTS | Atmospheric Chemistry Experiment Fourier Transform Spectrometer | PV4$\theta$ | 4.5-PVU dynamical tropopause-relative potential temperature |
| CARIBIC-2 | Civil Aircraft for the Regular Investigation of the atmosphere Based on an Instrument Container | PVU | Potential vorticity unit |
| DJF | December-January-February | RSTD | Relative standard deviation |
| EqL | Equivalent latitude | SON | September-October-November |
| IAGOS | In-service Aircraft for a Global Observing System | SPARC | Stratosphere troposphere Processes And their Role in Climate |
| JETPAC | JEt and Tropopause Products for Analysis and Characterization | START08 | Stratosphere-Troposphere Analyses of Regional Transport |
| JJA | June-July-August | STJ | Subtropical Jet |
| Lat | Latitude | STJ-L | STJ-relative latitude |
| MERRA-2 | Modern Era Retrospective-analysis for Research and Applications, version 2 | STJA | STJ-relative altitude |
| MLS | Microwave Limb Sounder | STJ$\theta$ | STJ-relative potential temperature |
| OCTAV | Observed Composition Trends And Variability | TACTS/ESMVAL | Transport and Composition in the Upper Troposphere and Lower Stratosphere and Earth System Model Validation |
| P | Pressure | $\theta$ | Potential temperature |
| PGS | POLSTRAC$^a$-GW-LCYCLE$^b$-SALSA$^c$ | UTLS | Upper troposphere/lower stratosphere |
| PV | potential vorticity | WMO | World meteorological organization |
| PV2A | 2-PVU dynamical tropopause-relative altitude | WMOA | Thermal tropopause-relative altitude |
| PV2$\theta$ | 2-PVU dynamical tropopause-relative potential temperature | WMO$\theta$ | Thermal tropopause-relative potential temperature |

$^a$Polar Stratosphere in a Changing Climate

$^b$Investigation of the Life cycle of gravity waves

$^c$Seasonality of Air mass transport and origin in the Lowermost Stratosphere

**Table A1.** Acronyms and symbols used in this study.



**B    Number of measurements per bin**



**Figure A1.** As Figure 2, but displaying the number of measurements (the count) in each bin for several coordinate systems.




# B    DJF variability for ACE-FTS and lidars



**Figure C1.** As Figure 8, but displaying the ACE-FTS ozone relative standard deviation.





**Figure C2.** As Figure 8, but displaying the lidar ozone relative standard deviation.



# D   JJA variability





**Figure D1.** As Figure 8, but displaying the MLS JJA ozone relative standard deviation.



**Figure D2.** As Figure 9, but displaying the ozonesonde JJA ozone relative standard deviation.



**Figure D3.** As Figure 10, but displaying the aircraft JJA ozone relative standard deviation.





**Figure D4.** As Figure C1, but displaying the ACE-FTS JJA ozone relative standard deviation.



**Figure D5.** As Figure C2, but displaying the lidar JJA ozone relative standard deviation.





# E   Variability with respect to EqL/$\theta$





**Figure E1.** As Figure 11, but in comparison to binning in Equivalent latitude and potential temperature.



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
