# Peer review of "Exploring ozone variability in the upper troposphere and lower stratosphere using dynamical coordinates"

_EGUsphere, 2024_

## Author Comment (AC1)

We thank the reviewer for his comments. Below are our responses in blue.

The main changes were:

- We added the main conclusion to the abstract
- We modified the symbol size for the ozonesondeds and lidar in figure 1
- We added a rationale for the period and datasets used.
- We added a rationale for the usage of "zonal" means.
- We added a brief discussion of the WMO multiple tropopauses and tropopause breaks
- We added several references as suggested by the reviewers.

**Review by Juan Antonio Añel**

Review of (egusphere-2024-144) by Millan et al.

In this paper, the authors present the analysis of several relative coordinate systems to define the transition layer between the troposphere and the stratosphere, dealing with the regimes in the upper troposphere, the tropopause itself, and the lowermost stratosphere. They use ozone concentrations as a key fingerprint.

First of all, I have co-authored some works with some of the authors of this manuscript. However, we have not collaborated over the last few years; therefore, I have not perceived a conflict of interest, and I think I can provide an objective review of this paper.

Also, I recommend citing several of my works here. I am not trying to impose their citations on the authors. I suggest them because I think they cover gaps in this case and will help create a more balanced, complete, and informative manuscript for the readers. I let the authors and the editor judge on it.

We have added some of the suggested references even though we believe that the review process is not meant to be a tool for the reviewers to bolster their citations.

My main comment is that the manuscript would benefit from explaining why this topic is relevant and explaining the potential applications of these coordinates.

We believe the introduction provides already details to explain why this topic is important; in particular, just the first sentence succinctly summarizes why it is critical to quantify UTLS composition variations. Extensive discussion of the applications of these coordinates is beyond the scope of this paper but many of the references included in the introduction (e.g., Hoor et al., 2004; Pan et al, 2004; Hegglin et al., 2009; Manney et al, 2011; Schwartz et al., 2015; Olsen et al., 2019) describe such applications and will help any interested reader learn more.

As a general comment, given that equivalent latitude is the best-performing coordinate system here and the resolution is 5 degrees (table 3), I think it is important to note that finer resolutions could improve the result using it. Also, the authors use the known piecewise-constant method to compute the equivalent latitude, which results could be improved up to an additional 5% using a Region of Interest technique (Añel et al. 2013). This could be noted in the Discussion or Summary.

Añel JA, Allen DR, Sáenz G, Gimeno L, de la Torre L (2013) Equivalent Latitude Computation Using Regions of Interest (ROI). PLoS ONE 8(9): e72970. https://doi.org/10.1371/journal.pone.0072970

(https://doi.org/10.1371/journal.pone.0072970).

For measurements with sparse coverage, a finer resolution in the Equivalent latitude bin size may not necessarily improve the results since there will be less measurement to average for a given bin. That said, we have added in the Discussion section the following sentence: Given the importance of equivalent latitude, other methods to calculate it (e.g., Añel et al 2013) could be explored in the future.

Line 24: multiple tropopause and tropopause fold conditions play an essential role, introducing uncertainty on whether the region is under tropospheric or stratospheric conditions. The authors should mention some relevant literature here: Randel et al. 2007, Añel et al. 2008, Wang and Polvani, Añel et al. 2013.

Randel, W. J., D. J. Seidel, and L. L. Pan (2007), Observational characteristics of double tropopauses, J. Geophys. Res., 112, D07309, doi:10.1029/2006JD007904.

Añel et al. (2008) Climatological features of global multiple tropopause events, J. Geophys. Res., 113, D00B08, doi:10.1029/2007JD009697.

Wang, S., and L. M. Polvani (2011), Double tropopause formation in idealized baroclinic life cycles: The key role of an initial tropopause inversion layer, J. Geophys. Res., 116, D05108, doi:10.1029/2010JD015118.

Añel, J.A., de la Torre, L. and Gimeno, L., 2012. On the origin of the air between multiple tropopauses at midlatitudes. T*he Scientific World Journal*, 2012. https://doi.org/10.1100/2012/191028

Dr Añel is correct in pointing out that multiple tropopauses and tropopause fold conditions play a role in driving the ozone variability, we have added some of the suggested references.

Line 25: it would be valuable to add information on the uncertainty by satellite measurements because of vertical resolution. The impact of the vertical resolution is implied in the sentence "Measurements available in this region are spatially and temporally limited …" The actual vertical resolution information is given in section 2.1.1, which describes the satellite measurements. We feel that it is not necessary to duplicate measurement details in the introduction.

Line 29: I would mention the cold-point tropopause (Gettelman et al., Pan et al. 2018) as it is relevant for water vapour and has been used many times instead of other definitions. Also, I suggest making it more explicit that Potential Vorticity (PV) can be used to distinguish between stratospheric air masses and tropospheric ones and track them, as they have very different values. They could add some examples, e.g. Chen et al. (2013).

A GETTELMAN, P.M. de F FORSTER, A Climatology of the Tropical Tropopause Layer, Journal of the Meteorological Society of Japan. Ser. II, 2002, Volume 80, Issue 4B, Pages 911-924

Pan, L. L., Honomichl, S. B., Bui, T. V., Thornberry, T., Rollins, A., Hintsa, E., & Jensen, E. J. (2018). Lapse rate or cold point: The tropical tropopause identified by in situ trace gas measurements. *Geophysical Research Letters*, 45, 10,756–10,763. https://doi.org/10.1029/2018GL079573

Chen X, Añel JA, Su Z, de la Torre L, Kelder H, van Peet J, et al. (2013) The Deep Atmospheric Boundary Layer and Its Significance to the Stratosphere and Troposphere Exchange over the Tibetan Plateau. PLoS ONE 8(2): e56909. https://doi.org/10.1371/journal.pone.0056909

Eventually, OCTAV-UTLS is planning to look at the impact of coordinate systems for water vapor; in those studies, we will certainly include the cold point tropopause. For ozone, the cold-point tropopause is not commonly used.

Line 60: I think it is important to mention that the coordinates and definitions are also relevant and depend on the different phenomena to study (this probably can be addressed in the paragraph I mentioned before on the applications of this work); in many cases, even more critical than "regional" features.

The sentence now reads: Thus, the coordinates that are most helpful to study geophysical and transport properties of the data may be different for different regions and/or phenomena that are of interest.

Fig. 1: The dots and squares to locate the ozosonde and lidar sites are too big to be informative. It would be good to have them in a smaller size.

Table 1 includes the lat lon info of the ozonesondes and lidars. Nevertheless, the figure was updated as suggested.

[Figure]

Line 90: I understand the reasons for it, but it would be good to add a line with the reasons to use Aura-MLS and ACE-FTS: lengthening the time series, measuring principle, etc.

The sentence was modified: In this study, we focus on two satellite limb sounders, Aura MLS and ACE-FTS, to exploit their long time-series and maximize the overlap with other datasets.

For example, I would move the current lines 108-110 here. We decided not to move that sentence, we want to introduce MLS before discussing how ACE-FTS compare to it.

Also, I have found it quite surprising that Toohey et al. (2013) and Hegglin and Tegtmeier (2017) are not cited in this subsection, as they directly discuss the bias in ozone measurements by the instruments used here, and several of the authors of this manuscript (and myself) are co-authors of both these works.

Characterizing sampling biases in the trace gas climatologies of the SPARC Data Initiative, J. Geophys. Res. Atmos., 118, 11,847–11,862, doi:10.1002/jgrd.50874.

Hegglin and Tegtmeier (2017) https://doi.org/10.3929/ethz-a-010863911.

*We added those citations after the sentence: "In comparison with MLS, ACE-FTS has much lower sampling density and thus shows a seasonally varying sampling bias"*

*Now it reads: In comparison with MLS, ACE-FTS has much lower sampling density and thus shows a seasonally varying sampling bias (Toohey et al., 2013; Millán et al., 2016; Hegglin and Tegtmeier, 2017).*

*Where Millán et al 2016 is: Millán, L. F., Livesey, N. J., Santee, M. L., Neu, J. L., Manney, G. L., and Fuller, R. A.: Case studies of the impact of orbital sampling on stratospheric trend detection and derivation of tropical vertical velocities: solar occultation vs. limb emission sounding, Atmos. Chem. Phys., 16, 11521–11534, https://doi.org/10.5194/acp-16-11521-2016, 2016.*

Line 158: the year for "Smit and Thompson" is missing.  *Fixed*

Line 161: It could seem evident that the 50 hPa region is outside the UTLS regime. Therefore, the results should be fine with the mentioned problem with the ozonesondes. However, it would be good to be clear with numbers about the reasons, mentioning that it is because the cases when the tropopause extends up to 50 hPa and above (double and triple tropopause cases reflecting a transition layer yet) are below the 5% for most of the planet and below 20% in only a few regions (which however coincide for example with the ozonesonde for Boulder). This information can be found in Añel et al. (2008)

*We meant that top limit of the plots shown in the manuscript was close to 50 hPa so the drop-offs occur at pressure levels not studied in this manuscript. The sentence was changed to: These drop-offs were typically limited to pressures above ~50 hPa, which is approximately the upper limit of the vertical range used in this study. Therefore, the results shown here should generally be unaffected.*

Line 231: VMR has not been defined before. *VMR was changed to volume mixing ratio*

Line 266. Rather than the studies cited, I would cite the primary studies dealing with the exposition of the multiple tropopause phenomenon (Randel et al. 2007 and Añel et al. 2008).

*We have added these citations*

Also, I understand that mentioning intrusions here is a generalization, which is not entirely accurate. MTs in this region are not necessarily associated only with intrusions understood

in the sense of vertical movement but also with the latitudinal mix and overlapping of the tropical tropopause over the extratropical one and undergo latitudinal advection. This is mentioned later in line 274, but it should be added here and clarified to avoid misinterpretations.

Since this is mentioned in the same paragraph, we don't think we need to clarify it. To add it in line 266 will break the flow and be repetitive.

Line 274: regarding the horizontal mixing, again cite Wang and Polvani (2011) and Añel et et al. (2012). We added the Wang and Polvani reference. Anel et al 2012 is published in, what many consider, a predatory journal (https://en.wikipedia.org/wiki/The_Scientific_World_Journal) and hence we decided not to include such reference.

From Fig. 6, it is clear that the 4.5 PVU value catches better than the 2.0 PVU value in the stratospheric character and does much better in extratropical regions. This is not new at all. Later in the text (in the Discussion), the authors mention that it matches previous findings by Kunz et al. (2011a); however, already a prior work by Hoinka (1998) made clear that values above 3.5 PVU are a better representation of the extratropical tropopause. The result again makes a point against the extended use of the 2 PVU value to define the tropopause, which is clearly an overestimation. This point could be included in the Discussion.

Hoinka, K. P., 1998: Statistics of the Global Tropopause Pressure. Mon. Wea. Rev., 126, 3303–3325, https://doi.org/10.1175/1520-0493(1998)126<3303:SOTGTP>2.0.CO;2.

We added that reference before the Kunz et al. (2011a) citation.

Lines 282-283: please do not use parenthesis this way. https://eos.org/opinions/parentheses-are-are-not-for-references-and-clarification-saving-space

The sentence in question was changed to: Compared to other datasets, MLS displays larger RSTD values in the Northern extratropics and smaller values in the Southern extratropics in the tropopause-based coordinates. Despite its coarse vertical resolution potentially failing to properly resolve the tropopause, this RSTD values might be related to its better coverage of the region, i.e., MLS might sample more variability.

Both the datasets and code for the analysis should be better deposited in long-term repositories with DOI (e.g., PANGAEA, Zenodo). I know it is not a journal requirement, but it is good practice for the assets that the authors can do with reasonable effort.

The ozone datasets are available elsewhere as specified in the data availability section. The dynamical diagnostics are only available upon demand. That said, we are currently working on translating the IDL JETPAC software into python so that it is truly helpful for the community, i.e., without the expensive license for IDL, and that software will be made public.

---

## Author Comment (AC2)

We thank the reviewer for his/her comments. Below are our responses in blue.

The main changes were:

- We added the main conclusion to the abstract
- We modified the symbol size for the ozonesondeds and lidar in figure 1
- We added a rationale for the period and datasets used.
- We added a rationale for the usage of "zonal" means.
- We added a brief discussion of the WMO multiple tropopauses and tropopause breaks
- We added several references as suggested by the reviewers.

**Review by Reviewer 1**

This study explores ozone variability in the upper troposphere and lower stratosphere using plenty of coordinates. The results show obvious differences in either ozone concentration its variability while choosing different coordinates. Such results give important hints in detecting changes in ozone on different time scales and in different regions, and support to the OCTAV-UTLS activity. The manuscript is well organized and written in English. I would recommend an acceptance after some minor revisions.

Comments:

Abstract: It would be better to summarize the main findings in this manuscript.

We added at the end of the abstract: Overall, the use of equivalent latitude-potential temperature leads to the most substantial reduction in binned variability across the UTLS. This coordinate pairing uses PV on isentropic surfaces thus following the transport of tracers in adiabatic frictionless flow.

1. L60-70: Here, the authors describe the 'geophysical variability' and its importance. However, it is not clear for how to distinguish the 'geophysical variability' and the true variability. In the analysis of this study, the authors evaluate different coordinates by comparing the relative standard deviation of ozone presented by different coordinates, but did not explain why reduced relative standard deviation is better. I think the relative standard deviation includes both 'geophysical variability' and the true variability, what is the scientific meaning of a reduced relative standard deviation?

The reviewer is correct. The relative standard deviation encompasses dynamical variability, atmospheric trace gas "chemical" variability, and 'geophysical noise'. The

purpose of reducing the relative standard deviation of the binned data is to minimize the geophysical noise contribution to study a more realistic representation of the dynamical and trace gas variability. Lines 75 – 79 of the original manuscript describe this (which has been slightly modified for clarity): *In other words, process-related coordinates can reduce binned variability **(i.e., reduce the contribution from the geophysical noise**), highlighting a more realistic representation of the geophysical and trace gas variability....*

2. Section 2.2.2: In this section, the authors want to examine the effects of different coordinate systems on the representation of geophysical variability. However, the descriptions to each figure are very simple. I would suggest the authors to describe in more details to help the audience to understand the 'geophysical variability'.

   In that section we introduce the coordinate systems using the mean values of ozone. In section 3, we discuss the RSTD (the variability) for the same coordinate systems in the same figure layouts. Thus, we don't think it is needed to add the discussion about the variability at this stage. That said, to introduce further the idea of the impacts of the different coordinate systems we added the following sentences:

   Figure 2 illustrates the redistribution of ozone across these three coordinates when plotted versus latitude as the horizontal coordinate. *While the ozone distributions share some broad similarities, notable differences are observed, showcasing the impacts of using different vertical coordinates. The impact of these coordinates on the ozone variability will be discussed in section 3.*

   An example of these relative coordinates is illustrated in Figure 3, which shows ozone plotted as a function of latitude and potential temperature relative to the three tropopauses used in this study. *Tropopause coordinates segregate measurements taken in the troposphere from those taken in the stratosphere, leading to strong gradients at the zero coordinate level (i.e., the tropopause). The usefulness of these coordinates in minimizing binned variability depends on how well the corresponding tropopause captures these ozone gradients, as well as the vertical resolution of the measurements in question.* The bounds of the vertical coordinate grids...

3. L274: it is evident that. Done

4. L398: the use of tropopause or subtropical jet 'vertical coordinates', should be horizontal coordinates? This sentence is confusing, please rewrite it.We want to

remind the reviewer that the subtropical jet can be used in the vertical, for example when referring to the altitude of the jet core, or in the horizontal, when referring to the latitude of the jet core. The sentence was changed to: Across all datasets, referring to the tropopause or STJ core in the vertical leads to greater binned variability in altitude based coordinates compared to potential temperature based coordinates, irrespective of the horizontal coordinate used.

---

## Author Comment (AC3)

We thank the reviewer for his/her comments. Below are our responses in blue.

The main changes were:

- We added the main conclusion to the abstract
- We modified the symbol size for the ozonesondeds and lidar in figure 1
- We added a rationale for the period and datasets used.
- We added a rationale for the usage of "zonal" means.
- We added a brief discussion of the WMO multiple tropopauses and tropopause breaks
- We added several references as suggested by the reviewers.

**Reviewer 4**

Review of « Exploring ozone variability in the upper troposphere and lower stratosphere using dynamical coordinates » by Millan et al., submitted to ACP

This manuscript aims to assess the usefulness of different transport-relevant coordinate systems (altitude, pressure, potential temperature, equivalent latitude, distance to the subtropical jet and distance to the tropopause) for dividing the measurements into bins affected by different atmospheric regimes. Then, the overall objective is to combine measurements from different platforms with different sampling characteristics for assessing the ozone trends and attributing them to changing atmospheric dynamics. This study is definitely an important milestone of the SPARC OCTAV-UTLS activity and follows a previous analysis by Millan et al, AMT 2023 (with almost the same co-authors), which was dedicated to the presentation of such dynamical diagnostics to describe the meteorological context for multi-decadal observations in the UTLS by ozonesondes, lidars, aircraft, and satellite.

The manuscript is well organized and well written. The figures are good and support the analysis. I recommend the publication after addressing some comments and suggestions, in order to improve its impact and make it useful for other data sets.

General comments :

In order to clarify some aspects of the methodology, of the results and therefore increase the impact (the usefulness) of these dynamical coordinates, I propose the following suggestions.

1. The major comment concerns the improvement in providing a further clarification of the objectives and of the results. This manuscript should better address the complementarity and/or the difference with the previous Millan et al., published in AMT in 2023, in the introduction.

After careful consideration, we have decided not to mention Millan et al., 2023 in the introduction. As correctly pointed out by the reviewer in the first paragraph of his/her review, this paper primarily focuses on the computation of dynamical diagnostics. In other words, Millan et al., 2023 is a technically oriented paper that describes the algorithms used to identify jets, tropopauses, and EqL at the measurement locations. Therefore, we believe it is more appropriate to mention it in section 2.2, where we describe the methods used to characterize the jets and tropopauses. The introduction primarily focuses on the importance of UTLS ozone and the introduction of coordinate systems. We feel that discussing the intricacies of computing dynamical diagnostics is not suitable for this section.

> At the end, the reader misses a clear opinion on the advantages of using these coordinates and a further understanding of the ozone variability in the UTLS, or at least a further discussion on the gain in consistency within the different data sets.

We believe the bullets in the summary section clearly summarize the impact on the relative standard deviation observed when using different coordinate systems. To further emphasize our main conclusion, we have added the following to the abstract (as suggested by multiple reviewers): "Overall, the use of equivalent latitude-potential temperature leads to the most substantial reduction in binned variability across the UTLS. This coordinate pairing uses PV on isentropic surfaces thus following the transport of tracers in adiabatic frictionless flow."

> It is quite frustrating to read that interesting results will be published in two future studies without giving more information in this one.

We understand the reviewer's frustration, however one paper cannot encompass all the material that OCTAV-UTLS will explore. The current manuscript already includes 11 figures in the main manuscript and 9 figures in the appendix, which we believe are essential to sustain our conclusions. Furthermore, many of these figures have up to 33 panels. We believe a longer paper that encompass other topics would lack a clear focus and would not do justice to the individual topics included.

> The last sentence of this manuscript "Another study will analyze how differences in sampling patterns and resolution (both vertical and horizontal) can affect the representation of the datasets as well as the trend quantification" is giving the negative impression that this manuscript is not going far enough to be really useful. It reveals that differences in sampling patterns and resolution are not addressed here. Therefore, the conclusions are somewhat weakened.

In the manuscript, we argue that despite variations in sampling and resolution across datasets, the conclusions drawn remain consistent across all datasets.  The physical reason for this is that assuming a tracer conservation on the time scale of jet- and tropopause dynamics in the UTLS, the tracer relationships on isentropes are conserved for adiabatic motions (namely, "This coordinate pairing uses PV on isentropic surfaces thus

following the transport of tracers in adiabatic frictionless flow."). Notably, this is independent from sampling density and location as long as adiabatic conditions are not violated.

Thus, these conclusions are robust since the consistency across the datasets (despite their differences) reinforces the validity and reliability of the findings.

The sentences that mention this in the manuscript are:

We examine the effects of different coordinate systems on the representation of geophysical variability in UTLS ozone through production of climatologies from the datasets outlined in Section 2.1. Because the variability in these climatologies is also influenced by sampling and measurement characteristics, the use of multiple datasets allows exploration of the commonalities among differences in climatologies as a function of coordinate system for each instrument. Any common changes between coordinate systems are assumed to result from a change in the representation of the effects of geophysical variability.

To further emphasize this point we added in the last paragraph of the manuscript:

In this study we identified coordinate systems that most help to reduce binned variability over broad regions in an effort to facilitate more robust UTLS composition trend analyses. **The use of multiple datasets with different sampling and resolutions enables us to identify commonalities among them, ensuring conclusions that are independent of the specific measurement techniques. We are aware that several questions regarding the binned variability are still open and some of them will be addressed in upcoming studies.** For example, a future OCTAV-UTLS study will …

2. Regarding the data sets used in this analysis, it is important to further argue about the selection of these data sets. Why are there not the same as in Millan et al., 2023 ? Why is IAGOS-CORE not used here in addition to IAGOS-CARIBIC ? Why is the number of ozonesondes so limited ? Why not use the ones from the SHADOZ network with the advantage of sampling the tropical regions ?

Although, we would ideally like to include all datasets available, the reality is that time, computing time, and even funding precludes us using all available ozone datasets. In fact, we considered using IAGOS-CORE in the beginning of the OCTAV-UTLS formulation, but due to the lack of resources we were not able to pursue it at that time. However, we are constantly increasing the datasets included in OCTAV. To address this, we have added at the end of the dataset section:

The datasets used in this study are not intended to be comprehensive; numerous other ozone records are available. For example, limb scattering satellite sounders, such as the

Optical Spectrograph and Infrared Imager System (OSIRIS; Llewellyn et al., 2004) or the Ozone Mapping and Profiler Suite (OMPS; Seftor et al., 2014), the long term airborne measurements from IAGOS-CORE (Petzold et al., 2015), and the ozonesondes included in the Southern Hemisphere ADditional OZonesondes (SHADOZ; Witte et al., 2017; Thompson et al., 2017). However, the records included are representative of the currently available measurement techniques in terms of resolution and geophysical sampling of the UTLS.

My suggestion would be to consider adding, as a result of this analysis, a list of recommendations for using such dynamical coordinates with other datasets. That would be valuable for the scientific community focused on providing ozone data sets and would increase the impact of this study.

To further emphasize the coordinate system that overall reduces the binned variability, we have added in the abstract: "Overall, the use of equivalent latitude-potential temperature leads to the most substantial reduction in binned variability across the UTLS. This coordinate pairing uses PV on isentropic surfaces thus following the transport of tracers in adiabatic frictionless flow."

We have also added in the summary section: "These conclusions were drawn using a variety of ozone measurements (i.e., ozonesondes, lidars, and satellite and in-situ aircraft measurements) with a plethora of vertical and horizontal resolutions, as well as sampling characteristics. Therefore, we anticipate that these results are applicable to other datasets not included in this study, such as OMPS, OSIRIS, IAGOS-CORE, and additional ozonesondes and lidar data available elsewhere."

3. Regarding the sampling patterns, the manuscript would be improved by adding a discussion on the impact (or not) of the differences in measurements locations. This comment is indeed linked to the one on the selection of the used data sets. MLS and ACE data sets are clearly "global" data sets but all the others are not and cannot be considered "symmetric zonally" like the satellite data sets. The sondes and lidars used in this study are only or mostly located in the "Western Hemisphere", on contrary of the CARIBIC aircraft data sets which spans a wider range of longitudes. What is the impact of such differences in discussing consistency in terms of zonal averages?

As mentioned in our response to question 1, we argue that despite variations in sampling and resolution across datasets, the conclusions drawn remain consistent across all datasets and hence they are robust through all available datasets.

Also a brief discussion or a simple pedagogic explanation on the use of zonal averages for presenting these transport-relevant coordinates would be a valuable addition to the manuscript. A few comments in the manuscript mention some characteristics varying with longitudes (e.g., double tropopause, strength and sharpness of the subtropical jet). These differences in the representativeness of the different data sets should be addressed or the differences (if any) between the two hemispheres (western vs eastern) should be discussed using the data sets providing the full range of longitudes. For example, Cohen et al., 2028, showed that the IAGOS-CORE data sets have different levels of ozone between the North America – Atlantic and the Eurasian sectors in the UTLS, when the tropopause is defined as the 2 PVU. Providing zonal averages have clear advantages, but when it comes to reducing and analysing the ozone variabilities, this longitudinal dependence has to be clearly discussed.

We have added in the Coordinate mapping section:

For this initial study, we use averages over all longitudes with different horizontal coordinates, similar to zonal means when using latitude. However, many dynamical and chemical processes exhibit significant longitudinal variations. Consequently, as mentioned in the introduction, coordinates that are most helpful to study geophysical and transport properties may vary depending on the region or phenomenon of interest.

4. Regarding the sampling period: What is the rationale to cover the 2005-2018 period while some of the data sets (i.e. aircraft) cover only a few years, and not the same for all (according to Table 1) ? A further discussion on this choice and on the impact (or not) of merging data sets from 2008 with those from 2015-2016 and those apparently equally distributed over the long 2005-2018 period would be valuable to add.

To clarify our rationale, we added at the beginning of section 2.2.2:

We chose this period due to the current availability of dynamical diagnostics (discussed in section 2.2.1), which require significant computing time to generate. This period allows for ample overlap among all measurement techniques used here, i.e., ozonesondes, lidars, aircraft in-situ campaigns, and limb sounders. While the aircraft in-situ measurements from PGS, TACTS/ESMVAL, and START08 do not cover the entire period, we include them to enhance the coverage of this measurement technique. However, it's worth noting that the bulk of the variability is driven, in the aircraft results, by the overwhelming quantity of CARIBIC-2 measurements.

5. Regarding the differences in the vertical resolution: What is the impact (or not) of different vertical resolutions, among the data sets themselves (i.e. from 3 km for MLS to 100 m for the ozonesondes and probably less for aircraft) and with the vertical spacing of theMERRA-2 products (1.2 km) ? I recommend that table 1 includes the information on the vertical resolution and for aircraft, the detailed "Range" instead of flight levels which is not very informative as research aircraft may fly higher or lower than passenger aircraft.

Again, we are precisely exploiting such differences to draw robust conclusions. As requested by the reviewer we tried to add the vertical resolution information to table 1 but were unsuccessful, there is not enough horizontal space to add another column. We have added the typical "range" of the flights as a note in table 1.

Specific comments:

1. The abstract should better highlight the main findings by adding a few sentences from the Summary section.

We added at the end of the abstract: Overall, the use of equivalent latitude-potential temperature leads to the most substantial reduction in binned variability across the UTLS. This coordinate pairing uses PV on isentropic surfaces thus following the transport of tracers in adiabatic frictionless flow.

2. Line 120 : A more general and recent reference to IAGOS should be added here, or at least the web site, e.g. htpp://www.iagos.org; Petzold, et al., 2015; Thouret et al., 2022

We have added the recommended references

3. Line 155 : Can you further explain this gridding ? Is such a 100 m gridding to reduce computing power applied to other data sets ? It is quite surprising as the ozone data set is probably not the "heaviest". In general, it would be nice to have the same types of details in all sections describing the different data sets.

When computing the dynamical diagnostics, the algorithm interpolates the reanalysis fields to the measurement locations. Many years ago, in one of the first OCTAV-UTLS meetings, it was decided that we were going to reduce the fine resolution measurements to avoid needless interpolation of the reanalysis fields (which typically have about 1-km vertical spacing in the UTLS) to very fine grids of, say, 5 meters (as the ozonesondes) where the information they supply would be redundant. We have modified the manuscript to:

In this study, ozonesondes were gridded to 100 m to reduce computing power when calculating the dynamical diagnostics (see Section 2.2). **It is important to note that this gridding resolution has no impact on the study's results, as the reanalysis fields only contain information at about 1-km vertical spacing and measurements will be averaged together in approximately 1-km bins.**

4. Line 158 : year is missing in the reference Smit and Thompson, as well as in the references list, line 607. It has been published in 2021.

Corrected

5. Lines 164-166 : The question is then "why not using more ozone sondes stations in this analysis ?"

We have added at the end of the dataset section:

The datasets used in this study are not intended to be comprehensive; numerous other ozone records are available. For example, limb scattering satellite sounders, such as the Optical Spectrograph and Infrared Imager System (OSIRIS; Llewellyn et al., 2004) or the Ozone Mapping and Profiler Suite (OMPS; Seftor et al., 2014), the long term airborne measurements from IAGOS-CORE (Petzold et al., 2015), and the ozonesondes included in the Southern Hemisphere ADditional OZonesondes (SHADOZ)(Witte et al., 2017; Thompson et al., 2017). However, the records included are representative of the currently available measurement techniques in terms of resolution and geophysical sampling of the UTLS.

Lines 204-206 : Further details are necessary here regarding the vertical resolutions, from the native ones to the gridded ones.

The native vertical resolutions are explained in the paragraphs corresponding to each dataset, below copied and pasted from the manuscript below:

Aura MLS: The MLS ozone vertical resolution in the UTLS is around ~ 3 km.

ACE-FTS: These measurements achieve an effective vertical resolution of around 1 km in the UTLS region due to vertical oversampling (Hegglin et al., 2008).

Airborne in-situ instruments: However, CARIBIC-2 aircraft operate at cruising altitudes of 10–13 km, near the climatological location of the extratropical tropopause. The high temporal and horizontal sampling of CARIBIC-2 provides a very detailed view of the tropopause and a very long time series (starting in 1997). In contrast, the other aircraft

missions studied here, START08, PGS, and TACTS/ESMVal, have more limited regional and temporal coverage, but provide more extensive vertical coverage of the UTLS, making them ideal for process-oriented studies.

Lidars: Most lidars achieve high vertical resolution, on the order of less than 1 km.

Ozonesondes: In this study, ozonesondes were gridded to 100 m to reduce computing power when calculating the dynamical diagnostics (see Section 2.2).

6. Line 220 : "than" should be replaced by "that".

Corrected

7. Line 223-224 : Would it be because the range of sampled longitudes with these data sets is restricted to the "western hemisphere" (se also General Comment #3) ?

This is just a consequence of the sparse sampling of the ozonesondes and lidars used, it has nothing to do with the actual ozonesonde or lidar technique. The paragraph explaining this has been left unchanged.

References:

Cohen, Y., Petetin, H., Thouret, V., Marécal, V., Josse, B., Clark, H., Sauvage, B., Fontaine, A., Athier, G., Blot, R., Boulanger, D., Cousin, J.-M., and Nédélec, P.: Climatology and long-term evolution of ozone and carbon monoxide in the upper troposphere–lower stratosphere (UTLS) at northern midlatitudes, as seen by IAGOS from 1995 to 2013, Atmos. Chem. Phys., 18, 5415–5453, https://doi.org/10.5194/acp-18-5415-2018, 2018.

Petzold, A., Thouret, V., Gerbig, C., Zahn, A., Brenninkmeijer, C.A.M., Gallagher, M., Hermann, M., Pontaud, M., Ziereis, H., Boulanger, D., Marshall, J., Nédélec, P., Smit, H.G.J., Friess, U., Flaud, J.-M., Wahner, A., Cammas, J.-P., Volz-Thomas, A. & Team, I. (2015). Global-scale atmosphere monitoring by in-service aircraft – current achievements and future prospects of the European Research Infrastructure IAGOS. (Vol. 67, pp. 28452). https://doi.org/10.3402/tellusb.v67.28452

Thouret, V., Clark, H., Petzold, A., Nédélec, P. & Zahn, A. (2022). IAGOS: Monitoring atmospheric composition for air quality and climate by passenger aircraft. (pp. 1-14). https://doi.org/10.1007/978-981-15-2527-8_57-1

---

## Author Comment (AC4)

We thank the reviewer for his/her comments. Below are our responses in blue.

The main changes were:

- We added the main conclusion to the abstract
- We modified the symbol size for the ozonesondeds and lidar in figure 1
- We added a rationale for the period and datasets used.
- We added a rationale for the usage of "zonal" means.
- We added a brief discussion of the WMO multiple tropopauses and tropopause breaks
- We added several references as suggested by the reviewers.

**Reviewer 3**

This is a well-focused, well-constructed, and well-executed study evaluating the use of various reference coordinates for examining upper troposphere lower stratosphere ozone distributions. Numerous datasets are leveraged to carry out the analysis and the results are consequently very robust. While it is lean on new discoveries, the study is nevertheless a worthwhile precursor to more extensive efforts expected in the future. I have nothing but a handful of minor suggested revisions below.

The Abstract: some text should be added to the abstract to capture the main conclusions of the work. As it stands now, the abstract is a bit too vague and descriptive of the effort rather than the outcomes. A synthesis of the bulleted items from Section 5 or at the very least the most important elements of them would suffice.

We added at the end of the abstract: Overall, the use of equivalent latitude-potential temperature leads to the most substantial reduction in binned variability across the UTLS. This coordinate pairing uses PV on isentropic surfaces thus following the transport of tracers in adiabatic frictionless flow.

In several places within Section 3 & 4, the text would benefit from a discussion of the tropopause break, tropopause errors, and some other common features. The WMO tropopause would be impacted most by some of these challenges and it is important to emphasize why such is problematic and why (physically) alternatives such as PV or simply potential temperature would/should/could perform better.

We changed the first sentence discussing the WMO versus PV tropopauses: In general, Figure 6 suggests that dynamical tropopause based coordinates resolve the ozone gradients across the tropopause region better than the WMO tropopause based coordinate. **This is not unexpected as the WMO tropopause results in breaks and multiple tropopauses between the tropics and the extratropics (e.g., Randel et al,**

**2007, Pan et al., 2009, Homeyer et al., 2010) rather than a continuous transition as provided by the dynamical (PV) tropopauses.**

Line 106: "effective resolution" should be "effective **vertical** resolution". Done

Line 114: Add "2008" before the open paren as there was also a START05 that preceded this mission. Done

Line 184: suggest adding Tian & Homeyer 2019 here too. Added

Line 331: recommend deleted "sub-" as "categories" seems sufficient. Done

Line 332: "coordinates" should be "coordinate". We changed the sentence from "the use of a tropopause-relative altitude coordinates" to "the use of tropopause-relative altitude coordinates" Fixing the plural error

---

## Author Response (AR2)

We thank the editor for his comments. In short, we have accepted all his recommendations. Below are our responses in blue.

**Comments by editor**

All 4 referees had positive views of your original submitted paper and my impression is that you have answered their comments thoroughly and used good judgement in deciding where revision is needed and where it is not. Therefore I will be pleased to accept your paper for publication in ACP without requesting further opinion from the referees.

We thank the editor for accepting our manuscript.

However I invite you to consider one point further before submitting a final version of the paper. Please provide a very brief response confirming whether you have or have not made the suggested changes.

You make the statement 'Hegglin et al. (2008) introduced the term "geophysical noise"' and (in your revised version) follow that a few sentences later with the sentence 'In other words, process-related coordinates can reduce binned variability (i.e., reduce the contribution from the "geophysical noise"), highlighting a more realistic representation of the geophysical and trace gas variability, and thus helping to elucidate the physical processes controlling it in different regions.'

I am bit concerned by (i) the use of the term "geophysical noise" and (ii) the use of "realistic" in the second sentence.

I thought that it was interesting that Reviewer 1, when referring to this part of the text, did not use the term "geophysical noise" but instead used "geophysical variability". Then when I looked at Hegglin et al (2008) I found that the term "geophysical noise" was used only in the abstract. I suggest that if you are going to use this term then you need to say what you mean by it and why you are using the term.

It seems to me that in other contexts "geophysical noise" is typically used to describe measurement uncertainty that is associated with uncertainty/variability in the geophysical background rather than, e.g. instrumental uncertainty. Typically this is because the

retrieval algorithm, converting the directly measured signal into the required estimate of a particular quantity, needs to make assumptions about the state of the geophysical background, which is unknown. In the context you are describing, your 'geophysical noise' is not limiting the accuracy of individual measurements -- e.g. of mixing ratio of a particular species -- but limits ability to take measurements and use them together to construct a useful picture of the overall state of the atmosphere. So my question is whether you really want to encourage the use of "geophysical noise" -- a term that was apparently used rather casually by Hegglin et al -- in this context.

We have removed all mentions of "geophysical noise" from the manuscript. The sentence discussed by the editor now reads:

For example, Hegglin et al. (2008) discussed this enhanced variability when comparing datasets binned using ...

Whether you do or not, I do recommend that you change the term 'realistic' in the second sentence to something else -- e.g. 'interpretable'. (Whilst I have no argument with the idea that the PV-theta binning of the observations is significantly helpful, I don't think that it adds realism.)

We changed realistic to interpretable

A further very minor suggested change:

Abstract: 'potential vorticity (PV)'.  Done